# Divalent metal cations stimulate skeleton interoception for new bone formation in mouse injury models

Wei Qiao[1,2,3,4,11], Dayu Pan[2,5,11], Yufeng Zheng [6], Shuilin Wu[7], Xuanyong Liu[8,9], Zhuofan Chen [10], Mei Wan [2], Shiqin Feng [5], Kenneth M. C. Cheung[1,3], Kelvin W. K. Yeung [1,3✉] & Xu Cao [2✉]

Bone formation induced by divalent metal cations has been widely reported; however, the underlying mechanism is unclear. Here we report that these cations stimulate skeleton interoception by promoting prostaglandin E2 secretion from macrophages. This immune response is accompanied by the sprouting and arborization of calcitonin gene-related polypeptide-$\alpha^+$ nerve fibers, which sense the inflammatory cue with $PGE_2$ receptor 4 and convey the interoceptive signals to the central nervous system. Activating skeleton interoception downregulates sympathetic tone for new bone formation. Moreover, either macrophage depletion or knockout of cyclooxygenase-2 in the macrophage abolishes divalent cation-induced skeleton interoception. Furthermore, sensory denervation or knockout of EP4 in the sensory nerves eliminates the osteogenic effects of divalent cations. Thus, our study reveals that divalent cations promote bone formation through the skeleton interoceptive circuit, a finding which could prompt the development of novel biomaterials to elicit the therapeutic power of these divalent cations.

[1] Department of Orthopaedics and Traumatology, Li Ka Shing Faculty of Medicine, The University of Hong Kong, Hong Kong S.A.R, People's Republic of China. [2] Department of Orthopaedic Surgery, The Johns Hopkins University School of Medicine, Baltimore, MD 21205, USA. [3] Shenzhen Key Laboratory for Innovative Technology in Orthopaedic Trauma, The University of Hong Kong-Shenzhen Hospital, Shenzhen 518053, People's Republic of China. [4] Applied Oral Sciences and Community Dental Care, Faculty of Dentistry, The University of Hong Kong, Hong Kong S.A.R, People's Republic of China. [5] Department of Orthopaedics, Tianjin Medical University General Hospital, 154 Anshan Road, Heping District, Tianjin 300052, People's Republic of China. [6] State Key Laboratory for Turbulence and Complex System and Department of Materials Science and Engineering, College of Engineering, Peking University, Beijing 100871, People's Republic of China. [7] School of Materials Science & Engineering, Tianjin University, Tianjin 300072, People's Republic of China. [8] State Key Laboratory of High Performance Ceramics and Superfine Microstructure, Shanghai Institute of Ceramics, Chinese Academy of Sciences, Shanghai 200050, People's Republic of China. [9] Cixi Center of Biomaterials Surface Engineering, Ningbo 315300, People's Republic of China. [10] Hospital of Stomatology, Sun Yat-sen University, Guangzhou 510000, People's Republic of China. [11] These authors contributed equally: Wei Qiao, Dayu Pan. ✉email: wkkyeung@hku.hk; xcao11@jhmi.edu

It has been known from the late 1990s that various divalent metal cations, such as magnesium ions (Mg$^{2+}$), zinc ions (Zn$^{2+}$), and copper ions (Cu$^{2+}$), play vital roles in bone growth, modeling, and remodeling[1–3]. Over the decades, the regulatory effects of these divalent cations on osteogenesis, osteoclastogenesis, angiogenesis, and immune responses have been gradually revealed[4–9]. However, it was not until recently that the involvement of nervous system in the new bone formation induced by divalent metal cations has begun to be realized[10]. So far, it remains unclear whether the central nervous system (CNS), which has emerged to play an important role in bone homeostasis[11–14], participate in this process. The CNS not only reacts to external stimuli, such as temperature, sound, odor, and taste, as exteroception but also receives signals from many physiological systems inside the body, including the cardiovascular, respiratory, gastrointestinal, genitourinary systems, and nociceptive systems, as interoception. In recent years, the interoceptive processes by which our body senses, interprets, integrates, and regulates signals from peripheral organs have emerged as a key mechanism for the control of internal state of our body by CNS[15]. The interocpetion system consists of ascending neural pathways that transmit the internal body signals to the brain, the CNS where the input interoceptive information is processed, and the descending neural pathways through which the interoceptive signals are circled back to regulate peripheral organs. In addition to the central and peripheral nervous system, several other components, such as the endocrine and immune systems, are proposed to be involved in the interoception process[16,17].

Unmyelinated free sensory nerve endings branch throughout tissues to recognize and carry signals related to pain, temperature, and mechanical stimuli[18]. As the largest organ, the skeleton accounts for more than 20% of human body weight. Bone tissues have abundant sensory and sympathetic innervations that connect bones with dorsal root ganglia (DRG) and the CNS[19–21]. The upregulation of sympathetic tone through serotonin and cAMP-response element binding protein (CREB) signaling in the hypothalamus contributes to decreased bone formation and increased bone resorption[13,22]. We have recently found that CNS senses bone density through prostaglandin E2 (PGE$_2$) as an ascending interoceptive signal and regulates bone formation through sympathetic nerves as the descending interoceptive pathway[14]. Specifically, PGE$_2$ secreted by osteoblasts in response to mechanical loading activates PGE$_2$ receptor4 (EP4) to stimulate phosphorylation of CREB in hypothalamus, where the tyrosine hydroxylase (TH) expression is downregulated for a sympathetic activity to induce commitment of mesenchymal stem/stromal cell (MSCs) to osteoblast lineage. Knockout of the EP4 gene in the sensory nerves or knockout of cyclooxygenase-2 (COX2) in the osteoblastic cells significantly increased sympathetic activity and reduced bone volume in adult mice. Our studies further demonstrate that ascending skeleton interoceptive signaling target hypothalamus to maintain the balance between bone and fat metabolism[23,24]. Moreover, we also showed sensory innervation into porous endplates is responsible for spinal hypersensitivity, thus, low-dose celecoxib maintains skeleton interoception in the endplate, thereby decreasing vertebral endplate porosity and innervation for the treatment of spinal pain[25,26]. Importantly, our most recent studies reveal that Skeleton interoception regulates bone and fat metabolism through hypothalamic neuroendocrine hypothalamic neuropeptide Y (NPY) (ref. [1]) and physiological PGE2 levels maintain skeleton interoception activity for bone homeostasis to reduce vertebral endplate porosity and spinal pain (ref. [2]). Therefore, the skeleton interoception could represent an essential circuit of the CNS in the control of bone metabolism and may shed a light on our understanding of divalent metal cation-induced bone formation.

Innate and adaptive immune processes, which become profoundly apparent after bone injury, play important roles in bone biology[27]. The presence of a series of immune cell-derived cytokines in the early inflammatory stage promotes the recruitment of fibroblasts, mesenchymal stem cells, and osteoprogenitor cells from their local niches to initiate bone repair[28,29]. For example, PGE$_2$, a well-recognized pro-inflammatory cytokine that is upregulated only during the initial stage of bone healing, is suggested to play an important role in new bone formation. The mutation of the 15-hydroxyprostaglandin dehydrogenase gene (HPGD), which is responsible for the degradation of PGE$_2$[30], plays a major role in promoting tissue regeneration[31,32]. Indeed, patients with an HPGD mutation have presented with subperiosteal new bone formation[33]. Since PGE$_2$ can elicit primary pain and prolong nociceptor sensitization[34,35], nonsteroidal anti-inflammatory drugs (NSAIDs), which inhibit cyclooxygenase (COX), the limiting enzyme of PGE$_2$, have been used to manage post-injury pain. However, there is increasing evidence that the use of NSAIDs, especially selective COX2 inhibitors, can affect bone healing[36,37]. These findings suggest that the inflammatory molecules produced by the immune system in response to bone injury may serve as biochemical signals to initiate the interoceptive control of bone regeneration.

In recent years, the immunomodulatory function of orthopedic biomaterials has been increasingly acknowledged. In fact, the host immune response to implanted biomaterials is now recognized as a determinant for the long-term survival and regenerative function of such biomaterials[38,39]. Owing to the central role of macrophages in the immune reaction to bone biomaterials, as well as their heterogeneity and plasticity, macrophage is one of the most important target cells for immunomodulation in the biomaterial field[39]. Divalent metal cations, such as Mg$^{2+}$, Zn$^{2+}$, and Cu$^{2+}$, have been extensively used to modify orthopedic biomaterials since the discovery of their osteogenic effects[8,40–43]. However, despite their well-recognized roles in the regulation of the immune response[44], the mechanism through which the divalent cation-modulated immune niche contributes to bone regeneration remains largely unclear. Indeed, the in vitro effects of divalent metal cations on osteogenesis starkly contradict our findings in a more complex in vivo model, because their osteogenic effects on the well-orchestrated bone healing process may involve the interplay of multiple systems in our body. We have recently showed the essential role of early inflammatory response in Mg$^{2+}$-induced new bone formation, as Mg$^{2+}$ stimulates macrophage via transient receptor potential cation channel member 7 (TRPM7) to create a pro-osteogenic immune microenvironment[7]. Since the effective window of Mg$^{2+}$ coincides with the reinnervation phase of bone healing[45], we hypothesize the communication between the immune system and the neural system can trigger the skeleton interoception for the regulation of new bone formation.

In this study, we sought to characterize the mechanism for divalent cation-induced bone formation. We found that divalent metal cations, including Mg$^{2+}$, Zn$^{2+}$, and Cu$^{2+}$, activated skeletal interoception through the immune-neural axis to initiate CNS regulation of bone formation. During the early stage of bone healing, divalent cation-induced PGE$_2$ secretion from macrophages stimulated EP4 in the sensory nerves. Importantly, the sprouting and arborization of sensory nerves in response to macrophage-derived PGE$_2$ transmitted interoceptive signals to the CNS to tune down sympathetic tone through hypothalamic CREB signaling, resulting in increased osteogenesis and decreased osteoclastogenesis in the injured bone. We revealed a previously

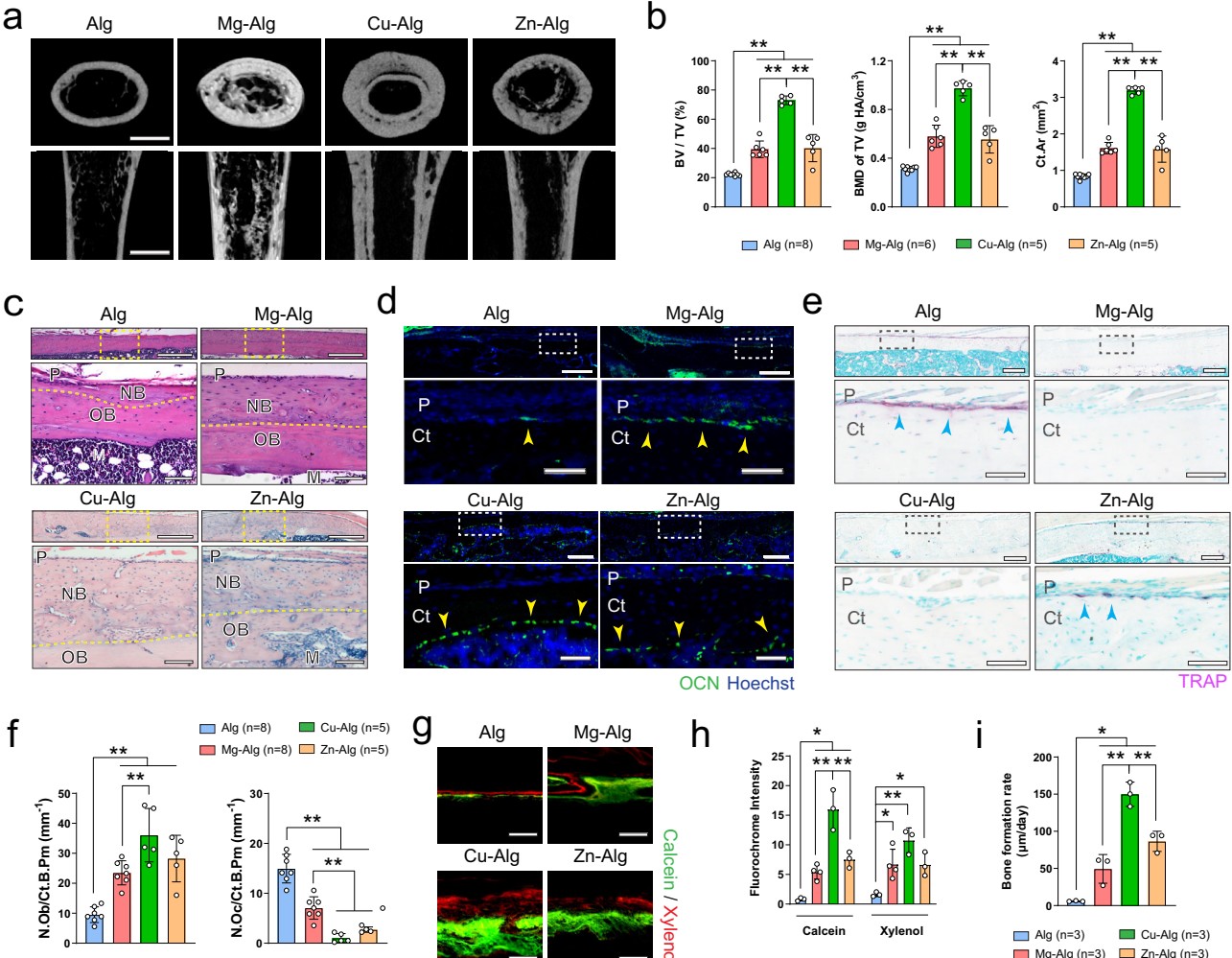

**Fig. 1 Divalent cations released from alginate induce new bone formation. a** Reconstructed micro-computed tomography (μCT) images (scale bars = 1 mm) showing the new bone formation in mouse femurs grafted with $Mg^{2+}$-, $Cu^{2+}$-, or $Zn^{2+}$-releasing alginate. Pure alginate grafted mice serve as control. **b** Corresponding measurements of bone volume fraction (BV/TV), bone mineral density (BMD of TV), and cortical bone area (Ct.Ar). **c** Representative images of hematoxylin and eosin (H&E) staining showing periosteal new bone formation ($n = 3$). Lower images (scale bars = 100 μm) are high-resolution versions of the boxed regions in the upper images (scale bars = 500 μm). P, periosteum; NB, new bone; OB, old bone; M, marrow. **d** Representative immunofluorescent images showing the presence of osteocalcin+ ($OCN^+$) osteoblasts on the cortical bone surface. Lower images (scale bars = 50 μm) are high-resolution versions of the boxed regions in the upper images (scale bars = 200 μm). P, periosteum; Ct, cortical bone. **e** Representative images of tartrate-resistant acid phosphatase (TRAP) staining showing the presence of $TRAP^+$ osteoclasts on the cortical bone surface. Lower images (scale bars = 100 μm) are high-resolution versions of the boxed regions in the upper images (scale bars = 400 μm). **f** Histomorphological analysis of osteoblast (N.Ob/Ct.B.Pm) and osteoclast (N.Oc/Ct.B.Pm) numbers on the cortical bone surface. Representative images of calcein/xylenol labeling showing periosteal new bone formation (**g**, scale bars = 100 μm), quantitative analysis of fluorescence intensity of calcein and xylenol (**h**), as well as corresponding measurement of bone formation rate (**i**). Data are mean ± standard deviation (s.d.) *P < 0.05, **P < 0.01 by 1-way analysis of variance (ANOVA) with Tukey's post hoc test. Source data are provided as a source data file.

unknown role of divalent cations in bone formation through skeletal interoception.

## Results

**Divalent cation-induced bone formation.** To investigate the effect of metallic divalent cations including $Mg^{2+}$, $Zn^{2+}$, or $Cu^{2+}$ on bone formation, we used an alginate-based hydrogel to facilitate the temporary and localized delivery of these divalent cations in mouse femurs (Supplementary Fig. 1a) after evaluating their cytotoxicity in vitro (Supplementary Fig. 1b). Pure alginate (Alg) or divalent cation-releasing alginate (i.e., Mg–Alg, Cu–Alg, and Zn–Alg) was placed in a tunnel defect drilled from the patello-femoral groove of the distal femur along the axis of the femoral

shaft. Micro-computed tomography (μCT) analysis showed a substantial increase in bone volume fraction (BV/TV), bone mineral density (BMD of TV), and polar moment of inertia (ρMOI) in the femur 4 weeks after the placement of divalent cation–releasing alginate compared with the pure alginate-treated control mice (Fig. 1a, b and Supplementary Fig. 2a). Particularly, cortical bone area (Ct.Ar), cortical bone thickness (Ct.T), and bone perimeter (B.Pm) were greater in the divalent cation-treated group compared with the controls (Fig. 1a, b and Supplementary Fig. 2a). We found a similar trend in the increase of trabecular thickness (Tb.T) in the femurs treated with $Mg^{2+}$ or $Cu^{2+}$, although it was less pronounced (Supplementary Fig. 2a). Indeed, hematoxylin and eosin (H&E) staining showed significant new bone formation located primarily at the peripheral cortex in the

divalent cation-treated femurs (Fig. 1c). Moreover, immuno-fluorescent staining revealed that the number of osteocalcin+ (OCN+) osteoblasts increased significantly on the endocortical and periosteal surfaces of femurs treated with $Mg^{2+}$, $Zn^{2+}$, or $Cu^{2+}$ (Fig. 1d, f). Meanwhile, the number of tartrate-resistant acid phosphatase+ (TRAP+) cells on the endocortical and periosteal surfaces decreased significantly in the divalent cation-treated femurs compared with the controls (Fig. 1e, f). In particular, the number of osterix+ and Runx2+ osteoprogenitors in the periosteum was also significantly higher after the release of $Mg^{2+}$ from the hydrogel (Supplementary Fig. 2b, c). Furthermore, using fluorochrome labeling, we showed that the release of divalent cations, including $Mg^{2+}$, $Zn^{2+}$, and $Cu^{2+}$, contributed to a significantly higher rate of mineral deposition, as manifested by an increased fluorochrome (i.e., calcein and xylenol) intensity and an increased distance between the two fluorochrome labels (Fig. 1g–i and Supplementary Fig. 2d). We also confirmed that the delivery of divalent cations didn't lead to any histological alteration in the spleen, liver, kidney, and heart tissues at either the early (week 1, Supplementary Fig. 3a) and later stage (week 4, Supplementary Fig. 3b) of the bone healing process. Therefore, our results reveal that divalent cations, including $Mg^{2+}$, $Zn^{2+}$, and $Cu^{2+}$, could stimulate periosteal new bone formation.

**Divalent cation-induced PGE2 production from macrophages.** To examine the mechanism of divalent cation-induced bone formation, we first characterized the immune response around the divalent cation-releasing alginate, because new bone formation in the periosteum can be evidenced by μCT and H&E staining (Supplementary Fig. 3a, b) at early inflammatory stage of bone healing (i.e., day 7 after the injury). The majority of immune cells present at this key stage of new bone formation were found to be F4/80+CD68+ macrophages rather than CD11c+ dendritic cells, CD19+ B-cells, or CD3+ T-cells (Supplementary Fig. 4c). We demonstrated that the number of CD68+ macrophages increased significantly in the bone marrow and periosteum of divalent cation-treated femurs compared with controls (Fig. 2a, b and Supplementary Fig. 5a). Co-immunostaining of COX2 with CD68 demonstrated that COX2 was expressed primarily in CD68+ macrophages in the periosteum (Fig. 2a), and that the stimulation of divalent cations, including $Mg^{2+}$, $Zn^{2+}$, or $Cu^{2+}$, significantly increased expression of COX2 in CD68+ macrophages in the callus (Fig. 2a, b and Supplementary Fig. 5b). Indeed, the release of $Mg^{2+}$ significantly elevated $PGE_2$ concentration in bone (Fig. 2c) and serum (Fig. 2d) during the early stage (i.e., week 1), when the injured site was undergoing acute inflammation. To examine the mechanism of divalent cation-induced COX2 expression, we isolated and cultured the primary mouse bone marrow macrophages (BMM) with the divalent cations. $Mg^{2+}$, $Zn^{2+}$, and $Cu^{2+}$ all promoted secretion of $PGE_2$ (Fig. 2e). Importantly, when used at a specific level (i.e., 0.1 mM $Zn^{2+}$, 0.1 mM $Cu^{2+}$, or 10 mM $Mg^{2+}$), the divalent cations upregulated prostaglandin E synthase (PTGES) gene expression (Fig. 2f) and protein levels of COX2 in the BMM (Fig. 2g) without affecting cell viability (Supplementary Fig. 1b). We also verified these findings by sorting macrophages in LysM-YFP mice for RT-qPCR assay after the placement of divalent cation-releasing alginate. Our data showed Mg–Alg, Cu–Alg, and Zn–Alg all contributed to significantly upregulated expression of PTGES and COX2 (Fig. 2h). Furthermore, divalent cations led to the phosphorylation of nuclear factor-κB (NF-κB) p65 and inhibitor of nuclear factor-κB (IκBα), the key cascade proteins in the NF-κB signaling pathway (Fig. 2i). Indeed, $Mg^{2+}$ significantly increased the binding of NF-κB p65 to the COX2 gene promoter as shown in chromatin immunoprecipitation (ChIP) assay (Fig. 2j). Thus,

we showed that divalent cations stimulate the secretion of $PGE_2$ from macrophages during the inflammation stage of the bone healing process.

**Ablation of COX2 in macrophages eliminates periosteal bone formation.** To validate the mechanism in vivo, we established an inducible macrophage-depleted mouse model by crossing LysM-Cre mice with $iDTR^{wt}$ mice. Macrophage depletion was effectively achieved in $iDTR_{LysM}^{+/-}$ mice by injecting diphtheria toxin (DTX). Macrophage depletion diminished the effect of $Mg^{2+}$ on bone healing, as there was no significant difference between the Mg–Alg group and the control group in bone formation on μCT (Fig. 3a, b and Supplementary Fig. 6a), whereas, $Mg^{2+}$-releasing hydrogel effectively promoted new bone formation after bone injury in LysM-Cre mice, similar to its effect in WT mice. Interestingly, when bone $PGE_2$ level was elevated with injection of SW03329 (a $PGE_2$ degradation enzyme inhibitor), the new bone formation in the injured femur improved significantly in both the control group and the Mg–Alg group (Fig. 3c, d and Supplementary Fig. 6b). Specifically, new bone formation at the peripheral cortex of the injured femur with injection of SW033291 in the control group was similar in the $Mg^{2+}$-treated group on H&E staining (Supplementary Fig. 6c). Moreover, the numbers of OCN+ osteoblasts and TRAP+ osteoclasts on the endocortical and periosteal surfaces of femurs were similar regardless of the presence of $Mg^{2+}$ in the alginate with injection of SW033291 (Supplementary Fig. 6d, e).

We further generated mice with conditional COX2 knockout in the macrophages ($COX2_{LysM}^{-/-}$) by crossing $COX2^{wt}$ mice with LysM-Cre mice. $Mg^{2+}$-releasing hydrogel failed to induce thickening of cortical bone in injured femurs of $COX2_{LysM}^{-/-}$ mice compared with $COX2^{wt}$ mice (Fig. 3e, f and Supplementary Fig. 7a). $Mg^{2+}$-induced new bone formation at the peripheral cortex of femurs in $COX2^{wt}$ mice was not seen in $COX2_{LysM}^{-/-}$ mice on H&E staining (Fig. 3g). Again, the effect of $Mg^{2+}$ on OCN+ osteoblasts and TRAP+ osteoclasts on both endocortical and periosteal surfaces was diminished in $COX2_{LysM}^{-/-}$ mice (Fig. 3g, i). These data show that COX2 in macrophages is responsible for $PGE_2$ production for divalent cation–induced bone formation.

**Sensory nerves are essential for divalent cation–induced bone formation.** Given that $PGE_2$ activates EP4 in the sensory nerve to maintain skeletal interoception activity, we examined whether sensory nerves are associated with divalent cation-induced bone formation. Prominent sprouting and arborization of calcitonin gene-related polypeptide-α+ (CGRP+) sensory nerve fibers coursing longitudinally over the outer periphery of the reactive callus were observed in immunostaining of periosteum (Fig. 4a). The intensity of CGRP+ nerve fibers in the periosteum was significantly increased with divalent cations (Fig. 4b). Importantly, terminal dendrites of CGRP+ sensory nerves were spatially associated with CD68+ macrophages in the reactive periosteum (Fig. 4c). Moreover, the expression of CGRP was significantly increased in the ipsilateral DRG of mice treated with divalent cations (Fig. 4d–f). Interestingly, CGRP+ dendrite sprouting observed primarily around COX2+ macrophages in the periosteum of $COX2^{wt}$ mice treated with divalent cations was missing from $COX2_{LysM}^{-/-}$ mice (Supplementary Fig. 7b). We then created a sensory denervation mouse model ($TrkA_{Avil}^{-/-}$) by crossing sensory nerve-specific Cre (Advillin-Cre) mice with nerve growth factor receptor tropomyosin receptor kinase A (TrkA) floxed ($TrkA^{wt}$) mice to confirm the essential role of sensory nerve in divalent cation-induced bone formation. Indeed, $Mg^{2+}$ significantly increased the density of CGRP+ sensory nerve

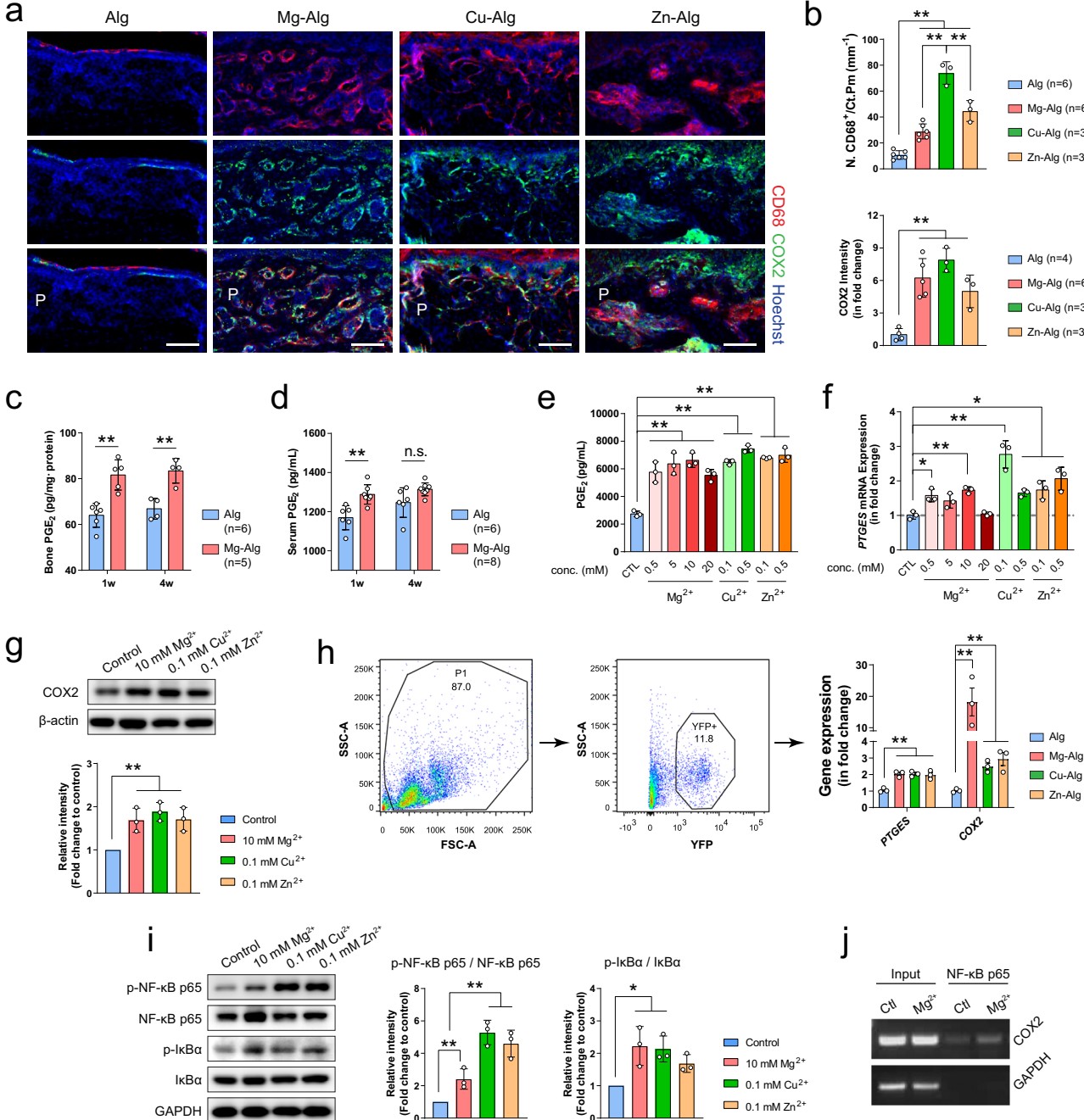

**Fig. 2 Divalent cations stimulate secretion of prostaglandin E2 (PGE₂) by macrophages. a** Co-immunostaining of cyclooxygenase-2 (COX2) with cluster of differentiation 68 (CD68) in the periosteum at week 1 postoperatively (scale bars = 100 μm). **b** Quantification of macrophage numbers on the cortical bone surface and COX2 intensity in the periosteum at week 1 postoperatively. Enzyme-linked immunosorbent assay (ELISA) analysis of the bone (**c**) and serum (**d**) $PGE_2$ concentration at week 1 and week 4 postoperatively. **e** ELISA analysis ($n = 3$) showing the effect of different concentrations of $Mg^{2+}$, $Zn^{2+}$, or $Cu^{2+}$ on the release of $PGE_2$ from bone marrow macrophages (BMM). **f** Real-time polymerase chain reaction analysis of $PTGES$ mRNA expression in mouse BMM after 3-day incubation culture medium supplemented with different concentrations of $Mg^{2+}$, $Zn^{2+}$, or $Cu^{2+}$ ($n = 3$). **g** Representative Western blots and corresponding quantification showing the effect of different divalent cations on the expression of COX2 in mouse BMM ($n = 3$). **h** Gating strategies for flow cytometry sorting of $YFP^+$ macrophages and the RT-qPCR data showing $PTGES$ and $COX2$ mRNA expression in macrophage at week 1 postoperatively ($n = 3$). **i** Representative Western blots and corresponding quantification showing the effect of different divalent cations on the phosphorylation of NF-κB and IκBα in mouse BMM ($n = 3$). **j** Chromatin immunoprecipitation assay showing that the NF-κB p65 at the COX2 promoter was upregulated by the stimulation of $Mg^{2+}$ ($n = 3$). Data are mean ± s.d. n.s. (nonsignificant), $*P < 0.05$, $**P < 0.01$ by one-way ANOVA with Tukey's post hoc test (**b**, **e**–**i**) or two-way ANOVA with Tukey's post hoc test (**c**, **d**). Source data are provided as a source data file.

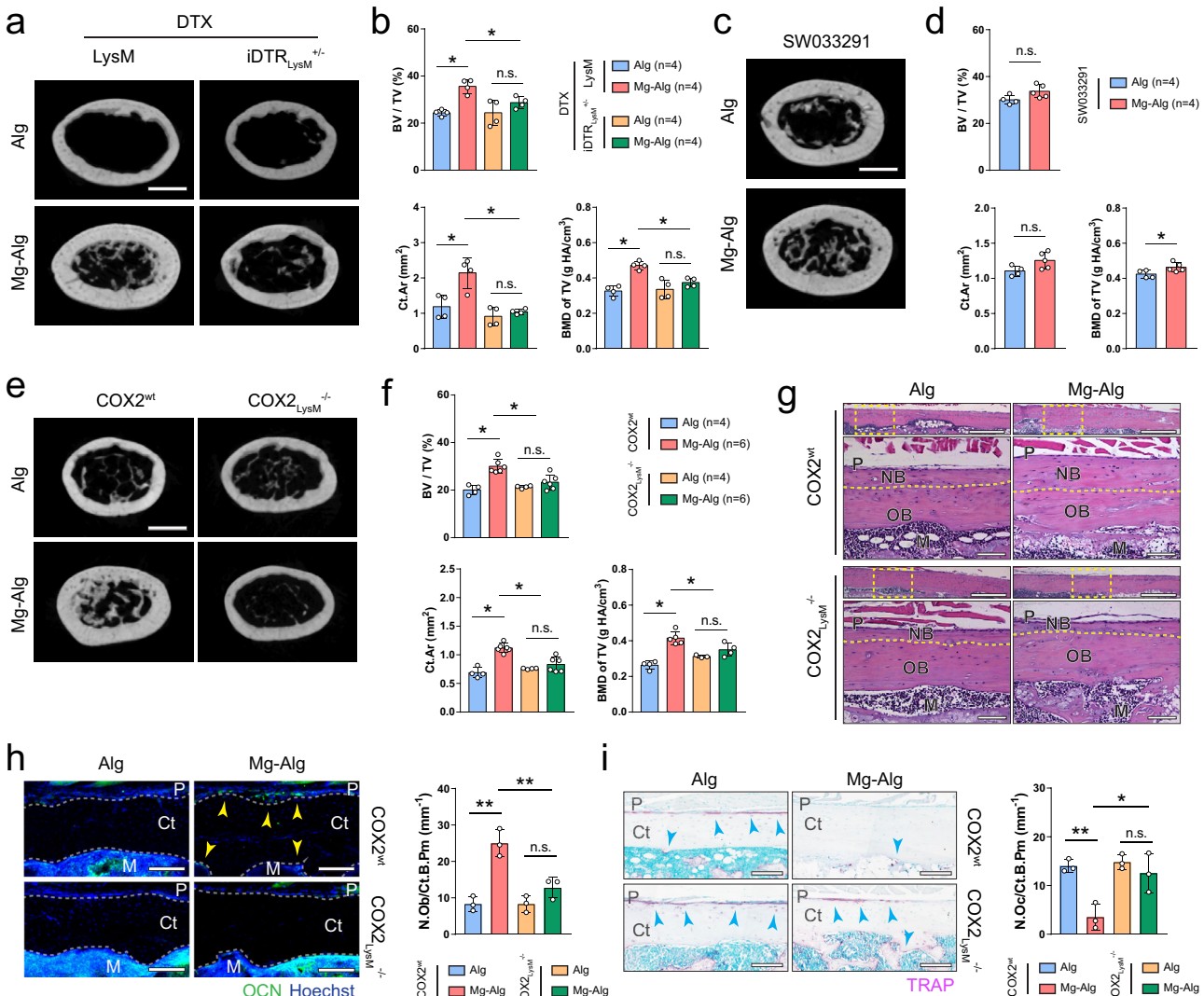

**Fig. 3 Macrophage-derived PGE₂ mediates divalent cation-induced bone formation.** Reconstructed μCT images (**a**, scale bars = 1mm) showing the cross-section of LysM or iDTR_LysM^+/− mouse femurs after the administration of diphtheria toxin and corresponding measurements (**b**) of BV/TV, BMD of TV, and Ct.Ar. Reconstructed μCT images (**c**, scale bars = 1mm) showing the cross-section of mouse femurs after the administration of SW033291 and corresponding measurements (**d**) of BV/TV, BMD of TV, and Ct.Ar. Reconstructed μCT images (**e**, scale bars = 1mm) showing the cross-section of femurs from COX2^wt mice or COX2_LysM^−/− mice grafted with pure alginate or Mg²⁺-releasing alginate, as well as corresponding measurements (**f**) of BV/TV, BMD of TV, and Ct.Ar. **g** Representative H&E staining images showing the periosteal new bone formation in COX2^wt or COX2_LysM^−/− mice (n = 3). Lower images (scale bars = 100 μm) are high-resolution versions of the boxed regions in the upper images (scale bars = 500 μm). **h** Representative immunofluorescent images and corresponding quantification showing the presence of OCN⁺ osteoblasts on the cortical bone surface (scale bars = 100 μm, n = 3). **i** Representative TRAP staining images and corresponding quantification showing the presence of TRAP⁺ osteoclasts on the cortical bone surface (scale bars = 100 μm, n = 3). Data are mean ± s.d., *P < 0.05, **P < 0.01 by one-way ANOVA with Tukey's post hoc test (**b**, **f**, **h**, **i**) or Student's two-sided T-test (**d**). Source data are provided as a source data file.

fibers in the bone marrow cavity and the endocortical and periosteal surfaces of TrkA^wt mice 7 days after surgery, whereas the number of CGRP⁺ sensory nerve fibers in TrkA_Avil^−/− mice was significantly lower relative to WT mice (Fig. 4g). Similarly, Mg²⁺ stimulated bone formation in TrkA^wt mice, and such effects were eliminated in TrkA_Avil^−/− mice (Fig. 4h, i and Supplementary Fig. 7c). Taken together, divalent cation-induced sensory innervation in injured bone tissue is essential for new bone formation.

**Divalent cations stimulate PGE₂/EP4 skeleton interoception in downregulation of sympathetic activity.** We have shown that PGE₂ activates sensory nerves via EP4 to induce phosphorylation of CREB in the VMH of the hypothalamus as a skeletal

interoception pathway[22,46]. We, therefore, investigated whether local delivery of divalent cations could trigger CREB phosphorylation in the hypothalamus. Immunostaining of brain sections showed that phosphorylation of CREB was significantly increased in the contralateral VMH of mice at 1 week after the delivery of Mg²⁺, Zn²⁺, or Cu²⁺ in the injured femurs (Fig. 5a, b). CREB phosphorylation in the hypothalamus with upregulation in 5-hydroxytryptamine receptor 2C (HTR2C) was confirmed by Western blot analysis (Fig. 5c, d). Moreover, using enzyme-linked immunosorbent assay (ELISA), we showed that Mg²⁺ significantly reduced epinephrine levels in serum and urine compared with controls, indicating suppression of sympathetic tone in Mg–Alg-injected mice (Fig. 5e). To examine whether divalent cation-induced CREB phosphorylation downregulates tyrosine

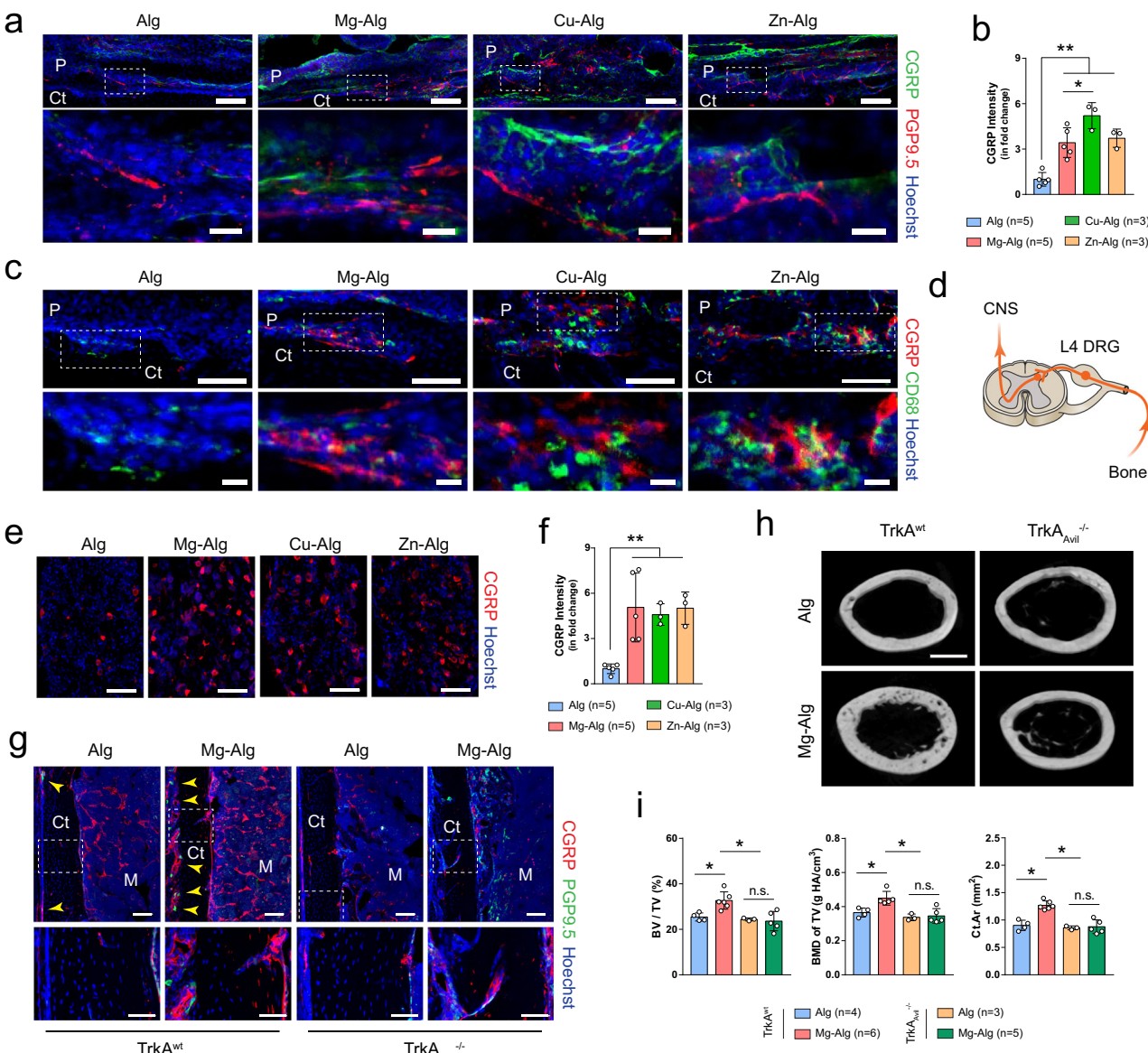

**Fig. 4 Deletion of sensory nerve abolishes divalent cation-induced bone formation. a, b** Representative immunofluorescent images and corresponding quantification showing the sprouting and arborization of calcitonin gene-related polypeptide-α$^+$ (CGRP$^+$) sensory nerves within the periosteum at week 1 postoperatively. Lower images (scale bars = 20 µm) are high-resolution versions of the boxed regions in the upper images (scale bars = 100 µm). **c** Co-immunostaining of CGRP with CD68 in the periosteum at week 1 postoperatively. Lower images (scale bars = 20 µm) are high-resolution versions of the boxed regions in the upper images (scale bars = 100 µm). DRG of L4 lumbar, which is responsible for the sensation of left femur, was harvested at week 1 postoperatively for histology study (**d**), representative immunofluorescent images (**e**), and corresponding quantification (**f**) showing the expression of CGRP (scale bars = 100 µm). **g** Representative immunofluorescent images showing the presence of CGRP$^+$ sensory nerves in the femurs of TrkA$^{wt}$ or TrkA$_{Avil}$$^{-/-}$ mice. Lower images (scale bars = 50 µm) are high-resolution versions of the boxed regions in the upper images (scale bars = 100 µm). Reconstructed µCT images (**h**, scale bars = 1 mm) showing the cross-section of femurs from TrkA$^{wt}$ mice or TrkA$_{Avil}$$^{-/-}$ mice grafted with pure alginate or Mg$^{2+}$-releasing alginate and corresponding measurements (**i**) of BV/TV, BMD of TV, and Ct.Ar. Data are mean ± s.d., *$P < 0.05$, **$P < 0.01$ by one-way ANOVA with Tukey's post hoc test. Source data are provided as a source data file.

hydroxylase (TH) for sympathetic activity, we performed immunostaining of TH for sympathetic nerves. The density of TH$^+$ sympathetic nerve fibers on the periosteal surface of divalent cation-treated femurs was significantly less in mice injected with divalent cation-releasing alginate compared with the controls (Fig. 5f). In parallel, spontaneous activity of mice, an indicator of sympathetic tone and postoperative pain[26], was assessed by spontaneous activity wheels. Compared with controls, the release of Mg$^{2+}$ contributed to significantly shorter daily distance and duration, as well as lower maximum speed of running-wheel activity at weeks 1 and 4 after surgery (Fig. 5g). Finally, we

examined the effect of divalent cations on the activation of skeletal interoception in TrkA$_{Avil}$$^{-/-}$ and COX2$_{LysM}$$^{-/-}$ mice, as well as their WT littermates. With sensory nerve denervation, the effect of Mg$^{2+}$ on phosphorylation of CREB was diminished compared with their WT littermates (Fig. 5h). Daily distance and duration of running-wheel activity in TrkA$_{Avil}$$^{-/-}$ mice were significantly longer than in their WT littermates (TrkA$^{wt}$), whereas such effect of Mg$^{2+}$ on decreasing spontaneous activity was abolished in TrkA$_{Avil}$$^{-/-}$ mice (Fig. 5i). Additionally, the increase of hypothalamic CREB phosphorylation triggered by Mg–Alg was abolished in COX2$_{LysM}$$^{-/-}$ mice (Fig. 5j). Similarly, the effect of Mg$^{2+}$

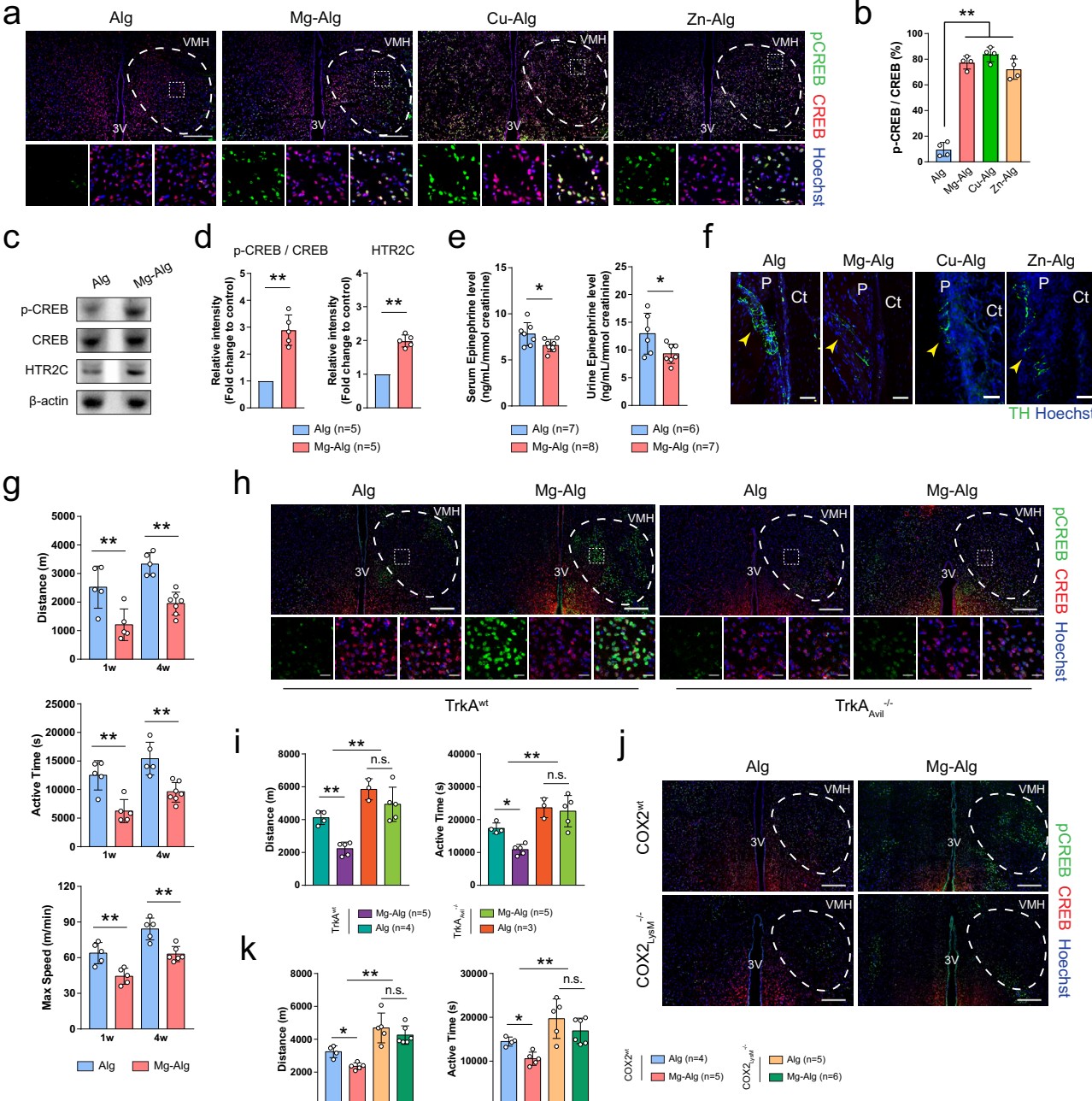

**Fig. 5 Divalent cations downregulate sympathetic activity through hypothalamic CREB signaling.** Representative immunofluorescent images (**a**) and corresponding quantification (**b**) showing the phosphorylation of CREB in the ventromedial hypothalamus (VMH) 1 week postoperatively ($n = 4$). Lower images (scale bars = 20 μm) are high-resolution versions of the boxed regions in the upper images (scale bars = 200 μm). Representative Western blots (**c**) and corresponding quantification (**d**) showing the phosphorylation of CREB and the expression of 5-hydroxytryptamine receptor 2C (HTR2C) in the hypothalamus tissue of mice at week 1 postoperatively. **e** ELISA of serum and urine epinephrine levels of mice at week 1 postoperatively. **f** Representative immunofluorescent images showing the presence of tyrosine-hydroxylase$^+$ (TH$^+$) sympathetic fibers on cortical bone surface ($n = 3$, scale bars = 50 μm). **g** The daily distance (m), duration (s), and maximum speed of running-wheel activity of WT mice at weeks 1 and 4 postoperatively ($n = 5$). Representative immunofluorescent images showing the phosphorylation of CREB in the VMH of TrkA$^{wt}$ or TrkA$_{Avil}$$^{-/-}$ mice (**h**), as well as COX2$^{wt}$ or COX2$_{LysM}$$^{-/-}$ mice (**j**) at week 1 postoperatively ($n = 3$). Lower images (scale bars = 20 μm) are high-resolution versions of VMH regions in the upper images (scale bars = 200 μm). The daily distance (m) and duration (s) of running-wheel activity of TrkA$^{wt}$ mice or TrkA$_{Avil}$$^{-/-}$ (**i**), as well as COX2$^{wt}$ or COX2$_{LysM}$$^{-/-}$ (**k**) mice at week 4 postoperatively. Data are mean ± s.d. *$P < 0.05$, **$P < 0.01$ by Student's two-sided T-test (**d**, **e**), two-way ANOVA with Tukey's post hoc test (**g**), or 1-way ANOVA with Tukey's post hoc test (**b**, **i**, **k**). Source data are provided as a source data file.

on decreasing the daily distance and duration of running-wheel activity was diminished in COX2$_{LysM}$$^{-/-}$ mice (Fig. 5k). Taken together, our data show that divalent cations promote PGE$_2$ from macrophages to activate skeletal interoception, resulting in downregulation of sympathetic tone and new bone formation.

**Knockout of EP4 in the sensory nerve inhibits divalent cation-induced activation of skeletal interoception.** EP4 is the PGE$_2$ receptor in the skeletal interoception and is known as the primary receptor in bone remodeling and homeostasis[47]. We first prepared conditional medium from Mg$^{2+}$-treated macrophages to

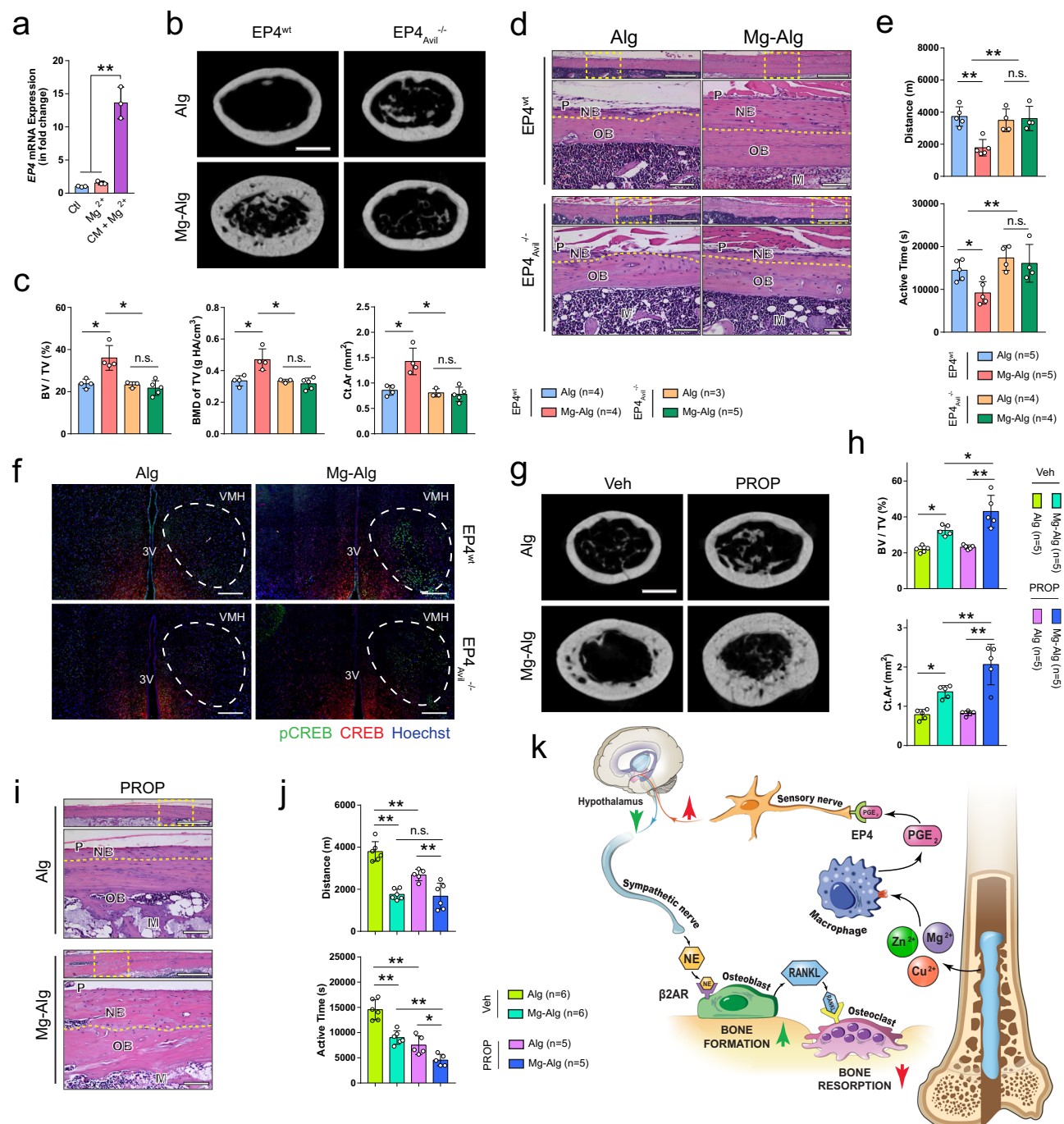

**Fig. 6 Knockout of PGE$_2$ receptor 4 (EP4) in sensory nerves abolishes divalent cation-induced bone formation. a** *EP4* expression in DRG neurons stimulated by 10 mM Mg$^{2+}$ with or without the conditioned medium from Mg$^{2+}$-treated macrophages ($n = 3$). Reconstructed μCT images (**b**, scale bars = 1 mm), corresponding measurements (**c**), and representative H&E staining images (**d**) showing the new bone formation in EP4$^{wt}$ mice or EP4$_{Avil}$$^{-/-}$ mice ($n = 3$). Lower images are high-resolution versions (scale bars = 100 μm) of the boxed regions in the upper images (scale bars = 500 μm). **e** The running-wheel activity of EP4$^{wt}$ or EP4$_{Avil}$$^{-/-}$ mice at week 4 postoperatively. **f** The phosphorylation of CREB in the VMH of EP4$^{wt}$ or EP4$_{Avil}$$^{-/-}$ mice at week 1 postoperatively ($n = 3$, scale bars = 200 μm). Reconstructed μCT images (**g**, scale bars = 1 mm), corresponding measurements (**h**), and representative H&E staining images (**i**) showing the new bone formation in mice injected with vehicle (Veh) or propranolol (PROP) Lower images (scale bars = 100 μm) are high-resolution versions of the boxed regions in the upper images (scale bars = 500 μm). **j** The running-wheel activity of mice injected with vehicle or propranolol at week 4 postoperatively. **k** Schematic diagram showing the findings of this study. The divalent cations triggered the production of PGE$_2$ from macrophages, which activated the EP4 at the sensory nerve to tune down sympathetic tones via the CREB signaling in the VMH, resulting in increased osteogenesis and decreased osteoclastogenesis in the periosteum. Data are mean ± s.d. *$P < 0.05$, **$P < 0.01$ by one-way ANOVA with Tukey's post hoc test (**a**, **c**, **e**, **h**, **j**). Source data are provided as a source data file.

test its effect on EP4 expression. EP4 expression in primary DRG neurons was increased more than 10-fold relative to treatment with $Mg^{2+}$ alone (Fig. 6a). We then generated $EP4_{Avil}{}^{-/-}$ mice by crossing Advillin-Cre mice with $EP4^{wt}$ mice. $Mg^{2+}$-induced bone formation was abolished in $EP4_{Avil}{}^{-/-}$ mice with induction of EP4 ablation in the sensory nerves, as shown by μCT (Fig. 6b, c and Supplementary Fig. 7d). We confirmed with H&E staining that the new bone formation induced by $Mg^{2+}$ at the peripheral cortex was diminished with conditional knockout of EP4 in sensory nerve fibers (Fig. 6d). Moreover, the increase of osteoblasts in the periosteum and decrease of osteoclasts were diminished in $EP4_{Avil}{}^{-/-}$ mice after postoperative administration of $Mg^{2+}$ (Supplementary Fig. 8a, b). Again, as in COX2 ablation mice, the effect of Mg–Alg on the daily distance and duration of running-wheel activity was also abolished in $EP4_{Avil}{}^{-/-}$ mice (Fig. 6e). Importantly, phosphorylation of CREB in the hypothalamus was not detectable in $EP4_{Avil}{}^{-/-}$ mice treated with Mg–Alg, unlike their $EP4_{Avil}{}^{-/-}$ littermates (Fig. 6f).

To validate the hypothesis that downregulation of sympathetic activity mediates $Mg^{2+}$-induced cortical bone thickening, we injected mice with propranolol, a widely used β2-adrenergic antagonist. Propranolol contributed to significantly more periosteal new bone formation in $Mg^{2+}$–Alg-injected femurs compared with the vehicle group, whereas it failed to increase bone formation in Alg-treated femurs relative to the vehicle group (Fig. 6g, h and Supplementary Fig. 8c). The bone that was newly formed in response to the stimulation of $Mg^{2+}$ after the administration of propranolol was located primarily at the peripheral cortex, as was previously observed in WT mice (Fig. 6i). Moreover, mice injected with propranolol had a shorter duration of running-wheel activity regardless of Alg or Mg–Alg treatment (Fig. 6j). Interestingly, the activation of CREB phosphorylation observed in the contralateral VMH was not prominent after the injection of SW033291 (Supplementary Fig. 8d), suggesting that the local stimulation caused by the $Mg^{2+}$-induced release of $PGE_2$ from macrophages was largely masked by the systematic administration of SW033291. Similarly, the decrease in spontaneous activity caused by Mg–Alg relative to Alg was diminished as the distance and duration of running-wheel activity in Alg treated mice were lower after the injection of SW033291 (Supplementary Fig. 8e). Therefore, EP4 in the sensory nerves mediates the activation of skeletal interoception to downregulate sympathetic activity, leading to new bone formation in the injured site.

## Discussion

In recent years, rapid progress has been made in the development of novel biodegradable metal implants[10,48]. With controlled degradation kinetics and gradual integration with bone tissue, biodegradable metal implants have been demonstrated to be superior to traditionally used bioinert metal implants for the treatment of musculoskeletal injuries. Particularly, these biodegradable implants can induce new bone formation through the release of various divalent metal cations[10,48]. Moreover, given the promising osteogenic properties of these divalent metal cations, they are also widely used for the modification of various kinds of orthopedic biomaterials[8,41,49]. Indeed, compared with expensive biological therapeutic agents and complex surgical procedures, the intervention using divalent cations appears to be a cost-effective way to achieve bone regeneration. However, it remains largely unclear how divalent cations induce new bone formation and whether these divalent cation-releasing biomaterials also induce bone formation through similar molecular and cellular mechanisms. In this study, we discovered that the divalent cations, $Mg^{2+}$, $Zn^{2+}$, and $Cu^{2+}$ serve as interoceptive signals to initiate CNS regulation of new bone formation after injury, which suggests the complexity of the underlying mechanism for divalent cation-induced bone regeneration has been greatly underestimated. Therefore, we believe that a clear understanding of the mechanism through which these divalent cations contribute to new bone formation through interoception will greatly benefit the development of novel biomaterials to fully elicit the therapeutic potential of these divalent cations.

The immune system and the nervous system are paramount to sense and respond to changes in our body, as they possess unique qualities that enable them to counter deviations in the internal environment[16]. Upon the delivery of divalent cations, which disrupts the homeostasis in bone microenvironment, the acute immune response contributes to a series of inflammatory chemokines and cytokines that relay information to the nervous system to initiate interoceptive control of bone formation. As a hallmark of inflammatory response after bone injury, COX2/$PGE_2$ plays a key role in cortical bone repair[33,36,37]. Subcutaneous administration of $PGE_2$ contributes to new bone formation on the endocortical and periosteal surfaces of ovariectomized and intact rats[50], whereas global knockout of COX2 almost eliminates of periosteal bone formation during bone healing[51]. In this study, we showed divalent cations, including $Mg^{2+}$, $Zn^{2+}$, and $Cu^{2+}$, significantly increased $PGE_2$ production from $CD68^+$ macrophages in the reactive periosteum. Interestingly, $PGE_2$ activated EP4 signaling in the $CGRP^+$ sensory nerve endings as the biochemical interoceptive signal. Moreover, our data showed that the injection of SW03329 elevated $PGE_2$ concentration in the control group to support cortical bone formation but did not further significantly promote $Mg^{2+}$-induced periosteal new bone formation. This implies that $PGE_2$-EP4 signaling in skeletal interoception is not entirely concentration-dependent, and there might be a specific window of $PGE_2$ levels for the activation of the immune-neural axis during bone healing—if so, this warrants further investigation. This observation suggests that secretion of $PGE_2$ specifically by macrophages stimulates divalent cations-induced bone formation through skeleton interoception.

Although the immunomodulatory effects of divalent cations have been reported, the inflammatory responses induced by these divalent cations don't always produce specific inflammatory cytokines (e.g., $PGE_2$) to facilitate new bone formation[52–55]. Instead, the immunomodulatory effects of these divalent cations are highly concentration-dependent and tissue-specific. Moreover, as bone tissue is the major reservoir for minerals in human body, the inflammation in this microenvironment induced by high levels of divalent cations may initiate the deposition of these trace minerals in hard tissue to alleviate their potential long-term toxicity. Therefore, the storage exogenous divalent cations in newly formed bone tissue could be an effective and efficient strategy to maintain the physiological condition in bone microenvironment. In this study, by utilizing the different cross-linking potential of $Mg^{2+}$, $Zn^{2+}$, and $Cu^{2+}$ on alginate[56], Mg–Alg, Zn–Alg, and Cu–Alg were designed exquisitely to harness the immunomodulatory effects of these divalent cations on macrophages. For instance, $Zn^{2+}$ and $Cu^{2+}$ released at a very low level (i.e., approximately 0.1 mM), which is also seen in either zinc or copper implants with slow degradation rate, produced $PGE_2$ without eliciting cytotoxicity. In comparison, 10 mM of $Mg^{2+}$ released from Mg-alginate, similar to that from degradable magnesium-based implants[57], stimulated the production of $PGE_2$ from macrophages, though $Mg^{2+}$ has been recognized as an anti-inflammatory agent due to its suppressive effects on pro-inflammatory molecules[58,59]. The immunomodulatory effects of the divalent cations were also shown manifested by the activation of the NF-κB signaling pathway in BMM. Given that $Mg^{2+}$ promotes the nuclear translocation of NF-κB p65 in binding to

$COX2$ promoter, it is anticipated that the NF-κB signaling pathway, which is crucial in inflammation responses[60], may play a central role in divalent cation-induced regulation of $PGE_2$. Although many of the inflammatory cytokines, such as IL-1β, IL-6, and TNF-α, can be recognized by their receptors expressed in the sensory nerve to modulate the nervous system[61], $PGE_2$, among all inflammatory mediators, tends to receive the most attention due to its role in mediating peripheral pain pathway[62]. By using a primary culture of DRG neurons and the mice with conditional knockout COX2 in macrophage, we demonstrated that the activation of sensory neurons is actually triggered by the macrophage-derived $PGE_2$ in response to the stimulation of divalent cations. Moreover, the ablation of EP4 in sensory nerves abolished the osteogenic effect of $Mg^{2+}$ in the periosteum, thus showing that the crosstalk between immune and neural system is essential in skeletal interoception-mediated new bone formation.

The density of sensory and sympathetic fibers in the periosteum remains the highest in the skeletal system[21]. The periosteum covers almost the entire bone surface and is one of the most regenerative tissues for skeletal osteogenesis[63]. Damage of the periosteum severely impairs cortical bone homeostasis[64,65] and bone fracture healing[10,65]. Moreover, an increase in the density of sensory and sympathetic nerve fibers in injured bone tissue has been reported to contribute to the bone healing process[45,66,67]. The sprouting and arborization of sensory nerve fibers in the periosteum in response to bone injury[45] or mechanical stimulation[68] may facilitate the sensation of stimuli and the initiation of osteogenesis. CGRP+ sensory nerves that emanate from the DRG of the spinal cord detect multiple stimuli in bone tissues (including inflammatory cytokines) and process and relay these signals to higher CNS levels[69]. In addition to its major role as a neurotransmitter and neuromodulator, CGRP is also considered a peptide that can be released from the peripheral nerve terminals to regulate osteoclast and osteoblasts[70,71], which was recently reported to be implicated in the new bone formation induced by pure magnesium implant[10]. In our study, a conspicuous increase in the number of CGRP+ nerve fibers in divalent cation-treated femurs suggests an association between the activation of sensory afferents and the inflammatory microenvironment, which is indispensable in skeleton interoceptive circuit. Therefore, the upregulation of CGRP in response to divalent cations may possess multiple functions, including the nociceptive transmission contributing to central sensitization and the direct control of bone cells through the receptors they shared. The spiral-like TH+ sympathetic nerve fibers typically wrap around the major blood vessels in the periosteum and penetrate the cortical bone via Volkmann's and Haversian canals[72,73]. They are known to be capable of regulating osteoblast activity through β2-adrenergic receptors (β2AR)[13,74]. Meanwhile, by increasing the secretion of the receptor activator of nuclear factor kappa-B ligand (Rankl) from osteoblasts, the sympathetic nervous system can also stimulate osteoclastic bone resorption[12,75]. We found that $PGE_2$ produced by divalent cation-stimulated macrophages tunes down sympathetic activity for the differentiation of OCN+ osteoblasts on the endocortical and periosteal surfaces. Moreover, in both the macrophage depletion and sensory denervation models, elevated $PGE_2$ in the periosteum failed to trigger the activation of CREB signaling in the VMH or periosteal new bone formation, indicating that the macrophage-mediated immune response and the sensation of afferent nerves are indispensable in the CNS recognition of stimulation from $Mg^{2+}$, $Zn^{2+}$, or $Cu^{2+}$ after bone injury. Importantly, CREB signaling in the ipsilateral DRG and the contralateral VMH by periosteal delivery of divalent cations downregulates sympathetic activity as a precise temporal-spatial feedback to the injured site for bone regeneration, as the

bone homeostasis of the contralateral femur was not altered after the activation of skeletal interoception (Supplementary Fig. 9a–c).

The discovery of the skeleton interoception-mediated new bone formation triggered by $Mg^{2+}$, $Zn^{2+}$, and $Cu^{2+}$ indicates that the complexity of the underlying mechanism for the osteogenic effects of these divalent cations has been greatly underestimated. Nevertheless, it is important to note that there have been multiple approaches reported in the past few decades through which these divalent cations may modulate bone homeostasis[6–9]. And this may explain the difference among the three tested divalent cations in term of their bone formation outcome. For instance, in our study, the highest bone volume was observed in the Cu–Alg group, as the old bone was barely resorbed from the endosteal surface by osteoclasts following the new bone formation on periosteal surface. In fact, from as early as 1981, $Cu^{2+}$ has been reported to have a direct dose-dependent inhibitory effect on osteoclastic activity[76]. $Cu^{2+}$ has also been used in biomaterials to shift the equilibrium between bone formation and bone resorption because the tolerance of osteoclasts and osteoblasts to $Cu^{2+}$ differs[77,78]. In addition to the bone cells, some divalent cations like $Mg^{2+}$ and $Zn^{2+}$ have also been shown to facilitate type H vessel formation by targeting endothelial cells, which couples angiogenesis with osteogenesis during bone healing[6]. This subtype of vessels, characterized by the co-expression of CD31 and endomucin, can provide niche signals for perivascular osteoprogenitors to promote osteogenesis[79,80]. Recent studies showed the sensory innervation following bone injury is an essential upstream mediator for vasculature[45], and electrical stimulation at DRG could promote type-H vessel formation to enhance bone regeneration[81]. Therefore, it would be interesting to further explore to which extend are angiogenesis and vasculogenesis involved in skeleton interoception and what their specific role may be in divalent cation-induced new bone formation.

In summary, we found that skeletal interoception mediates divalent cation-induced bone formation (Fig. 6K). The controlled delivery of $Mg^{2+}$, $Zn^{2+}$, or $Cu^{2+}$ stimulates to the production of $PGE_2$ from CD68+ macrophages, which is followed by rapid reinnervation of CGRP+ sensory nerve fibers in the reactive periosteum. Activation of EP4 receptor in sensory nerves by local $PGE_2$ induces CREB signaling in the VMH as an ascending interoceptive signal, which downregulates sympathetic activity as a descending interoceptive signal to induce osteogenesis at the injured bone treated with divalent cations. The discovery of divalent cation-induced bone formation through skeletal interoception could revolutionize the current understanding of bone regeneration and inspire innovative orthopedic biomaterials for bone tissue engineering.

## Methods

**Mouse breeding**. All animal experimental protocols and relevant ethical regulations were followed, and the study was approved by the Animal Care and Use Committee of The Johns Hopkins University, Baltimore, MD, USA (Protocol number: MO21M276). The *Advillin-Cre (Avil-Cre)* mouse strain was kindly provided by Xingzhong Dong (Department of Neuroscience, The Johns Hopkins University, Baltimore, MD, USA). The *TrkA[fl/fl]* mice were obtained from David D. Ginty (Department of Neurobiology, Harvard Medical School, Boston, MD, USA). The *LysM-Cre* mice and *iDTR[fl/fl]* mice were purchased from the Jackson Laboratory (Bar Harbor, ME, USA). The *COX2[fl/fl]* mice were provided by Harvey Herschman (Department of Biological Chemistry, University of California, Los Angeles, Los Angeles, CA, USA). The *EP4[fl/fl]* mice were obtained from Brian L. Kelsall (Laboratory of Molecular Immunology, National Institutes of Health, Bethesda, MD, USA). The Rosa26YFP reporter mice were obtained from Center for Comparative Medicine Research (CCMR), the University of Hong Kong. Heterozygous male *Avil-Cre* mice were crossed with a female *TrkA[fl/fl]* or *EP4[fl/fl]* mouse. The offspring were intercrossed to generate the following genotypes: wild type (referred to as WT in the text), *Avil-Cre* (Cre recombinase expressed driven by advillin promoter), *TrkA[fl/fl]* (mice homozygous for TrkA flox allele, referred to as

TrkA$^{wt}$ in the text), EP4$^{fl/fl}$ (mice homozygous for EP4 flox allele, referred to as EP4$^{wt}$ in the text), Avil-Cre::TrkA$^{fl/fl}$ (conditional deletion of TrkA receptor in Advillin lineage cells, referred to as TrkA$_{Avil}$$^{-/-}$ in the text), Avil-Cre::EP4$^{fl/fl}$ (conditional deletion of EP4 receptor in Advillin lineage cells, referred to as EP4$_{Avil}$$^{-/-}$ in the text). Heterozygous male LysM-Cre mice were crossed with a female iDTR$^{fl/fl}$ mouse or a COX2$^{fl/fl}$ mouse. The offspring were intercrossed to generate the following genotypes: WT, LysM-Cre, iDTR$^{fl/fl}$, COX2$^{fl/fl}$ mice (mice homozygous for COX2 flox allele, referred to as COX2$^{wt}$ in the text), LysM-Cre::iDTR$^{fl/-}$ (referred to as iDTR$_{LysM}$$^{+/-}$ in the text), LysM-Cre::COX2$^{fl/fl}$ (conditional deletion of COX2 in monocyte-macrophage lineage, referred to as COX2$_{LysM}$$^{-/-}$ in the text). Homozygous male LysM-Cre mice were crossed with a female Rosa26YFP mouse to generate LysM-YFP mice.

The genotypes of the mice were determined by polymerase chain reaction (PCR) analyses of the genomic DNA, which was extracted from mouse tails. The primers used for genotyping were Avil-Cre: Forward: CCCTGTTCACTGTGAGTAGG, Reverse: GCGATCCCTG AACATGTCCATC; LysM-Cre: Forward: CCCAGAAATGCCAGATTACG, Reverse: CTTGGGCTG CCAGAATTTCTC; TrkA loxP allele: Forward: AACAGTTTTGAGCATTTTCTA TTGTTT, Reverse: CAAAGAAAACAGAAGAAAAAT AATAC; iDTR loxP allele: Forward: GCGAAGAGTTTGTCCTCAACC, Reverse: AAAGTCGCTCT GAGTT GTTAT; COX2 loxP allele: Forward: AATTACTGCTGAAGCCCAACC, Reverse: GAATCTC CTAGAACTGACTGG; EP4 loxP allele: Forward: TCTGTGAAGCGA GTCCTTAGGCT, Reverse: CG CACTCTCTCTCTCCCAAGGAA. All animals were maintained at the animal facility of The Johns Hopkins University School of Medicine.

**In vivo treatment**. Twenty percent alginate gel (Alg) was prepared by mixing sodium alginate powder (Sigma-Aldrich, St. Louis, MO, USA,180947) in deionized water, while 10% magnesium chloride (MgCl$_2$, Sigma-Aldrich, M8266), 10% zinc chloride (ZnCl$_2$, Sigma-Aldrich, 208086), and 10% copper chloride (CuCl$_2$, Sigma-Aldrich, 307483) were used for the preparation of magnesium cross-linked alginate (Mg–Alg), zinc cross-linked alginate (Zn–Alg), and copper cross-linked alginate (Cu–Alg). Twelve-week-old male mice were anesthetized by intraperitoneal injection with ketamine (Vetalar, Ketaset, Ketalar; 75 mg/kg, intraperitoneally) and xylazine (Rompun, Sedazine, AnaSed; 10 mg/kg, intraperitoneally). A longitudinal incision was made at the left knee and the patella was dislocated to expose the femoral condyle. Using a 20-gauge needle (Becton, Dickinson and Company, BD Syringe, 309644), we created a tunnel with a diameter of 1 mm from the patello-femoral groove of the distal femur along the axis of the femoral shaft. After thorough irrigation with saline (Quality Biological Inc, 114055101), either pure alginate or divalent cation-releasing alginate (0.01 mL) was injected into the femoral canal. Meanwhile, a same tunnel defect was created in the right femur, but no material was injected. The wounds were sutured layer-by-layer, and the mice were housed in a specific-pathogen-free facility after the surgery. A monocyte-macrophage lineage depletion mouse model was induced by intraperitoneal injection of diphtheria toxin (DTX, Sigma-Aldrich, D0564) in iDTR$_{LysM}$$^{+/-}$ mice every other day during the week before and the week after the surgery. A PGE$_2$ degradation enzyme inhibitor, SW033291 (Selleck Chemicals, Houston, TX, USA), was administered by intraperitoneal injection at 10 mg/kg every day for 1 week after the surgery. A low-dose β-adrenergic receptor blocker, propranolol (PROP, Sigma-Aldrich, 1576005), was administered by intraperitoneal injection at 0.5 mg/kg every day for 1 week after the surgery.

**μCT analysis**. At designated time points, the femurs were harvested from the mice and fixed overnight using 4% paraformaldehyde. Analysis was performed using a high-resolution μCT scanner (SkyScan 1275, Bruker, Kontich, Belgium). The voltage of the scanning procedure was 65 kv with a 153-μA current. The resolution was set to 8.7 μm per pixel. Two phantom-contained rods with a standard density of 0.25 and 0.75 g/cm$^3$ were scanned with each sample for calibration. Data reconstruction was completed using NRecon software (v1.6, SkyScan), data analysis was accomplished using CTAn software (v1.9, SkyScan), and 3D model visualization was performed using CTvox software (v3.2, SkyScan). Bone volume fraction (BV/TV), bone mineral density (BMD of TV), trabecular number (Tb. N), trabecular thickness (Tb. Th), cortical thickness (Ct. Th), cortical area (Ct. Ar), bone perimeter (B. Pm), and ρ-moment of inertia (ρMOI) were measured via μCT data.

**Fluorochrome labeling**. Two fluorochrome labels were used sequentially to evaluate periosteal new bone formation rate after the placement of divalent cation releasing alginate. In brief, calcein green (5 mg/kg, Sigma-Aldrich, C0875) was subcutaneously injected into mouse femora one week after the surgery, while xylenol orange (90 mg/kg, Sigma-Aldrich, 52097) was injected two weeks after the surgery. The fluorochrome labels were visualized under a fluorescence microscopy (Niko ECL IPSE 80i, Japan). The intensity of fluorescence and the distance between two labeling were measured and analyzed by ImageJ software (NIH, USA).

**Immunofluorescence and histomorphometric analysis**. At designated time points, mice femurs were fixed with 4% paraformaldehyde (Fisher Scientific, SF100-4) overnight and decalcified with 10% ethylenediaminetetraacetic acid (pH = 7.4) for 21 days. For immunostaining, the samples were dehydrated in 20%

sucrose solution with 2% polyvinylpyrrolidone (PVP, Sigma-Aldrich, PVP40) for 24 h and embedded in 8% gelatin (Sigma-Aldrich, 1288485) supplemented with 20% sucrose (Fisher Scientific, S5-500) and 2% PVP. 40 μm-thick coronal-oriented sections of the femurs were obtained using a cryostat microtome. For histomorphometry, the samples were dehydrated in ethanol, embedded in paraffin, and prepared into 5 μm-thick coronal-oriented sections using a rotary microtome. The brain and DRG tissues harvested from the mice were fixed with 4% paraformaldehyde, dehydrated with 30% sucrose, and embedded in an optimal cutting temperature compound (OCT, Sakura Finetek, Torrance, CA, USA, 4583). 10 μm-thick coronal-oriented sections of the brain and DRG were obtained using a cryostat microtome.

Immunostaining was performed using a standard protocol. Briefly, the sections of the brain and DRG were incubated with primary antibodies to mouse OCN (Abcam, Cambridge, UK, ab93876, 1:200), CD68 (Abcam, ab31630, 1:400), CGRP (Abcam, ab81887, 1:300), PGP9.5 (Abcam, ab108986, 1:300), COX2 (Abcam, ab15191, 1:200), CREB (Abcam, 178322, 1:200), p-CREB (Abcam, ab32096, 1:200), and TH (Millipore Sigma, Burlington, MA, USA, AB152, 1:100) overnight at 4 °C. Alexa-Fluor 488-conjugated and Alexa-Fluor 647-conjugated secondary antibodies (Thermo Fisher Scientific, Waltham, MA, USA) were used for immunofluorescent staining, while the nuclei were counterstained with Hoechst 33324 (Thermo Fisher Scientific). Immunofluorescent images were captured using an LSM 780 confocal microscope (Zeiss, Oberkochen, Germany). For hematoxylin and eosin (H&E) staining, selected slides were stained in hematoxylin (Thermo Fisher Scientific, 7231) for 3 min and counterstained with eosin (Thermo Fisher Scientific, 7111) for 1 min. TRAP staining (Sigma-Aldrich, 387A) was performed in selected slides from each sample according to the manufacturer's instructions. In brief, selected slides were pre-warmed to 37 °C in waterbath and incubated in TRAP Staining Solution Mix 37 °C at for 15 min. The nucleus was counterstained by Methyl green (Millipore Sigma, M8884). Images were captured using a polarized light microscope (Nikon Eclipse VL100POL, Tokyo, Japan), and quantitative histomorphometric analysis was performed using Image J software (v.1.5, National Institutes of Health, Bethesda, MD, USA).

**Cell sorting**. All cell sorting was performed on a BD FACSAria$^{TM}$ SORP Cell Sorter (BD Biosciences, USA). At week 1 after the placement of divalent cation-releasing alginate, the femurs were harvested and rinsed with 1X PBS (Thermo Fisher Scientific, 10010023). After the removal of muscle tissue, bone marrow was flushed out with 1X PBS. The cells on the cortical bone were isolated by digesting the crushed bone chips in PBS containing 1 mg/mL Collagenase I (Sigma-Aldrich, SCR103) for 20 min at 37 °C. Cells were thoroughly washed, underwent red blood cell lysis (Invitrogen, Thermo Fisher Scientific, 00-4333-57), and resuspended in PBS containing 2% fetal bovine serum (FBS, Invitrogen, Thermo Fisher Scientific, 26140079). For cell sorting, YFP positive cells were sorted into chilled 15 mL centrifuge tubes containing PBS supplemented with 2% FBS. The harvested cells were washed and centrifuged before the addition of RNeasy lysis buffer (Qiagen, Germantown, MD, USA, 79216) for the extraction of total RNA.

**Cell culture**. Primary BMM and DRG neurons from 4-week-old mice were isolated. For the isolation of BMM, the mice were euthanized and both femurs were dissected to remove soft tissue. The femurs were then crushed into pieces and digested with α-MEM (Minimum Essential Medium Eagle α Modification, Fisher Scientific, MT10022CV) containing 3 mg/mL Collagenase I (Worthington Biochemical Corp, Lakewood, NJ, USA, LS004194), 4 mg/mL dispase (Sigma-Aldrich, D4818), and 1 U/mL deoxyribonuclease-I (Invitrogen, Thermo Fisher Scientific, 18047019). The single-cell suspension was achieved by passing the solution through a cell strainer to remove tissue fragments. After incubation in a humidified incubator with 5% CO$_2$ at 37 °C, the non-adherent cells were harvested and cultured in α-MEM supplemented with 20 ng/mL macrophage colony-stimulating factor (M-CSF, R&D Systems, Minneapolis, MN, USA, 416-ML) for 7 days. For the isolation of DRG neurons, DRGs from the L2–L5 spinal levels were isolated in cold DMEM/F12 (Dulbecco's Modified Eagle Medium/Nutrient Mixture F-12) medium (Invitrogen, Thermo Fisher Scientific, 11320033) and then digested with 1 mg/mL collagenase type A (Roche, Basel, Switzerland, 05349907103) at 37 °C. After trituration and centrifugation, cells were resuspended and seeded on glass coverslips coated with ploy-D-lysine and laminin. The culture medium was replaced 6 h after seeding, and the adherent cells were further cultured at 37 °C with 5% CO$_2$ for 3 days before use.

**ELISA**. Whole blood samples were collected by cardiac puncture immediately after the mice were euthanized. Serum was collected by centrifuging at $2000 \times g$ for 15 min and then stored at −80 °C before analysis. The total bone protein was harvested from the femurs grafted with pure alginate or Mg-Alg. The mid-shaft of the femur, approximately 1 cm long, was ground into mud using a ceramic mortar and pestle under cooling. The mud of the bone tissue was then homogenized in pre-cooled radioimmunoprecipitation assay (RIPA) lysis and extraction buffer (Thermo Fisher Scientific, 89900) for 1 h. The buffer solution was centrifuged at $15,000 \times g$ for 20 min at 4 °C. The supernatant was collected for protein concentration quantification with the BCA Protein Assay Kit (Thermo Fisher Scientific, A53225). An equal amount of protein from each sample was subjected to

quantitative analysis using a specific ELISA kit per the manufacturer's instruction. The $PGE_2$ concentrations in the serum and bone marrow were determined by the $PGE_2$ ELISA kit (Cayman Chemical, Ann Arbor, MI, USA, 514010). The OCN level was determined by an OCN ELISA kit (Biomedical Technologies Inc, Tewksbury, MA, USA, BT470). The serum and urine epinephrine levels were determined by an epinephrine ELISA kit (ALPCO, Salem, NH, USA, 17-EPIHU-E01.1).

**Quantitative real-time polymerase chain reaction (qPCR)**. The total RNA of the cells was extracted and purified using the RNeasy Plus kit (Qiagen, Germantown, MD, USA, 74034) per the manufacturer's instructions. For the reverse transcription, complementary DNA was synthesized using the SuperScript First-Strand Synthesis System (Invitrogen, Thermo Fisher Scientific, 18091050). The primers used in the RT–qPCR assay were synthesized by Life Technologies (Thermo Fisher Scientific) based on sequences retrieved from Primer Bank (http://pga.mgh.harvard.edu/primerbank, Supplementary Table 1). SYBR Green-Master Mix (Qiagen, Germantown, MD, USA, A46113) was used for the amplification and detection of complementary DNA on a C1000 Thermal Cycler (Bio-Rad Laboratories, Hercules, CA). The mean cycle threshold (Ct) value of each target gene was normalized to the housekeeping gene glyceraldehyde-3-phosphate dehydrogenase (*GAPDH*). The results were shown as a fold change using the ΔΔCt method.

**Western blot**. The total protein from animal tissues or cell cultures was lysed using RIPA lysis and extraction buffer supplemented with a protease inhibitor cocktail (Thermo Fisher Scientific, 78430). After centrifugation at $15,000 \times g$ for 10 min at 4 °C, the supernatants were collected to measure the protein concentration with the BCA Protein Assay Kit. A total of 30 μg of protein was subjected to sodium dodecyl sulfate-polyacrylamide gel electrophoresis and then blotted on the nitrocellulose membranes (Bio-Rad Laboratories, 1620115). The membrane was blocked in 5% w/v bovine serum albumin (BSA, Sigma-Aldrich, A9418) and incubated with blocking buffer-diluted primary antibodies overnight at 4 °C. The primary antibodies used were CREB (Abcam, 178322, 1:1000), p-CREB (Abcam, ab32096, 1:1000), HTR2C (ab197776, 1:1000), COX2 (Abcam, ab15191, 1:1000), p-IκBα (CST, 2859, 1:1000), IκBα (CST, 4814, 1:1000), p-NF-κB p65 (CST, 3033, 1:1000), NF-κB p65 (CST, 8242, 1:1000), and β-actin (CST,8457, 1:2000). The proteins were visualized by an enhanced chemiluminescence kit (Thermo Fisher Scientific, 34580) and exposed under a ChemiDoc XRS System (Bio-Rad Laboratories).

**Behavioral analysis**. The spontaneous activity of mice after surgery was assessed using spontaneous activity wheels (BIO-ACTIVW-M, Bioseb, Boulogne, France). Mice were housed in polycarbonate cages with free access to stainless steel activity wheels (diameter 23 cm; width 5 cm), which were connected to an analyzer that automatically recorded the distance traveled, mean speed, maximum speed, and total active time. The mice had *ad libitum* access to food and water during the test. They were allowed to acclimatize to the environment for at least 24 h before data were recorded.

**Chromatin immunoprecipitation (ChIP) assay**. After the stimulation, the ChIP assay was performed using an Agarose ChIP Kit (Thermo Fisher Scientific, 26156) according to the manufacturer's instructions. In brief, the chromatin was cross-linked by 1% formaldehyde and digested by micrococcal nuclease. The lysate was incubated with rabbit anti-NF-κB p65 at 4 °C overnight followed by incubation with ChIP Grade Protein A/G Plus Agarose. The purified DNA was analyzed by PCR assay using primers targeting mouse COX2 promoters: sense 5′-CCCGGAG GGTAGTTCCATGAAAGACTTCAAC-3′ and antisense 5′-GGTGGAGCTGGC AGGATGCAGTCCTG-3′. The primers targeting the *GAPDH* promoter served as a positive control. PCR products obtained after 40 cycles were separated on 2% agarose gels.

**Statistical analysis**. All data analyses were performed and illustrated using Prism software (v. 7, GraphPad Software, San Diego, CA, USA). Data are presented as means ± standard deviations (SD). For comparisons between two groups, two-tailed Student's T-tests were used. For comparisons among multiple groups, one-way or two-way analysis of variance (ANOVA) was used, followed by Tukey's post hoc test. Significant differences among groups were defined and noted as $*P < 0.05$ or $**P < 0.01$. The sample size was based on preliminary data, as well as on observed effect sizes.

**Reporting summary**. Further information on research design is available in the Nature Research Reporting Summary linked to this article.

## Data availability
All relevant data that support the findings of this study are available within the Article and Supplementary Information or from the corresponding author upon reasonable request. A Source Data file is provided with this paper. Source data are provided with this paper.

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

## Acknowledgements

We acknowledge the HKU Li Ka Shing Faculty of Medicine Faculty Core Facility for providing a harmonious working environment. This work is supported by Hong Kong Research Grant Council (#17214516, 17207719, K.W.K.Y.), Hong Kong Innovation Technology Fund (ITS/405/18, K.W.K.Y.), FOX Gift and Necrosis Fund (X.C.). For their editorial assistance, we thank Jenni Weems, MS, Kerry Kennedy, BA, and Rachel Box, MS, in the Editorial Services group of the Johns Hopkins Department of Orthopaedic Surgery.

## Author contributions

W.Q. and D.P. contributed equally to this work. W.Q. performed animal surgery, cell culture, and the in vitro and in vivo tests. D.P. contributed to the animal study and data analysis. Y.Z., S.W., and X.L. provide insightful comments on the material-science-related issues. K.M.C.C. and Z.F.C. contributed to the design of animal models and provided invaluable suggestions about the clinical indications of the study. M.W. and S.F. contributed to the experiment design and data interpretation. X.C. and K.W.K.Y. contributed to data interpretation and supervised the project. W.Q. illustrated the schematic diagram. W.Q., X.C., and K.W.K.Y. wrote the manuscript with input from all authors.

## Competing interests

The authors declare no competing interests.

## Additional information

**Peer review information** *Nature Communications* thanks Katharina Schmidt-Bleek, Mone Zaidi and the other anonymous reviewer(s) for their contribution to the peer review this work. Peer reviewer reports are available.

