## [Peer Review File · Nature Communications]

Reviewers' Comments:

Reviewer #1:

Remarks to the Author:

Review Manuskript NCOMMS-21-28944

Divalent Metal Cations Stimulate Skeleton Interoception to Promote Bone Formation

The manuscript describes the interconnectivity of divalent cations, bone formation, macrophages, periosteum and neural signalling. There are numerous aspects that have been addressed and analysed using alginate doped with cations administered to the bone marrow cavity of mice. WT mice and several specific mouse models are used to address specific questions. This is a very elaborate and complex study.

This manuscript addresses a highly complex topic and the introductions gives the reader a concise overview of the different topics: - introducing interoception, - the importance of neural signalling for bone, - impact of prostacyclin on bone; - divalent cations in biomaterials. The authors refer to a number of previous publications from their group as background to this study – perhaps a graphical abstract of all the here introduced principles underlying the current work would help the reader to grasp the complexity without feeling compelled to study further papers for an introduction to the topic.

The paragraph on the systemic effects on the brain resulting from the cations delivered locally raises the question whether the effects seen in the injured bone are local or if the bone homeostasis per se is altered in these animals. This raises the question of the dosage of cations applied in this experiments with respect to gaining better bone healing – how could that be translated towards a clinical application?

The connection between the cations and the macrophages is well founded by the presented data – but are there other immune cells or other cells effected by the cations – could the authors elaborate whether they have looked into other immune cells that could be involved?

The reported effect on the nervous system is raising the question whether the supply of these ions effects signalling pathways per se as they are known as essential co-factors important for enzyme functions. Did the authors look into systemic effects in the mice following the cation administration?

Within the manuscript the authors mention immunomodulation – but they only elaborate on PGE (as a pro-inflammatory signal) and on macrophages (as immune cells) – therefore only a very specific part of the immune reaction is being analysed – no further mention is made on the differences of the cations on the immune reaction following bone injury per se and their effect on the signalling cascade in bone is included in the manuscript. Please consider reformulating this aspect in the manuscript.

This also is true for biomaterials that contain cations that are released over time – are they really immunomodulatory implants – or is this more of a side effect? And there is again the question of the dosage – how is a cation release from implants comparable to the dosages used in this experimental setting.

Minor remarks:

For the test of injecting alginate and alginate with cations into the mouse femur, please include information on the volume of alginate that was injected. Please state how the concentration of cations (10%) was chosen – as the biocompatibility of the ions is varying (as shown in extended Fig1b) – please indicate why no evaluation of the best concentration for the optimal effect was necessary for the in vivo procedure.

The cation release kinetic showed that earlier time points coincide with the highest release – why did the authors chose to evaluate changes in bone after 4 weeks – how are they correlating the changes with the ion release and exclude secondary effects from participating in the resulting changes (eg. changes in the vasculature...) – in this respect please check the last sentence of the first results paragraph –“our results reveal that divalent.....”

Fig 2:

d) PGE concentration, please indicate where this is measured?

Fig 4: a and c – the images are so small, that it is difficult to “see” the nerve fibers and their connection with CD68 + cells – could they be enlarged?

Please check the following sentence: “Interestingly, CGRP+ dendrite sprouting located primarily around COX2+ macrophages in the periosteum of COX2wt mice treated with divalent cations was significantly reduced in COX2LysM^{-/-} mice (Extended data Fig. 5a). Line 177.

Reviewer #2:

Remarks to the Author:

This manuscript aims at defining the mechanism underlying a positive activity of metallic divalent cations on bone mass in a murine injury model. It shows that the presence of these ions triggers the synthesis of PGE₂ by bone marrow macrophages and an increase in CGRP levels in bone of wild-type but not Cox-2 knockout mice. It then uses sensory nerves depletion of TrkA and EP4 to show that these structures are relaying the signal to the hypothalamus to impact the sympathetic regulation of bone remodeling. This is a study that address a poorly understood observation in bone biology and it should be of general interest. The paper is generally well written but there are some technical concerns, outlined below, that should be carefully addressed before publication.

- The title should specify that the findings relate to an injury model in the mouse.
- To demonstrate de facto that the presence of divalent cations increases bone formation quantification of the bone formation rate following fluorescent labeling should be shown at least in figures 1, 3.
- The quantification of osteoblast and osteoclast numbers should be shown in the main figure 1 instead of the extended data. These data should also be shown for the experiments presented in figure 3.
- In figure 1, 2 and extended figure 3 the immunofluorescence data have not been quantified, only representative fields are shown. Unless a proper quantification is performed, the corresponding text should not refer to “significant” changes in expression.
- At least some of the in vitro data shown in figure 2f-h should be confirmed using cells directly isolated from the implanted mice.
- The western blots for changes in CREB/P-CREB and HT2C hypothalamic levels should be shown in figure 5. The number of mice used for these experiment should be clarified.
- The changes in CREB/P-CREB signal shown in figure 5a are not convincing. This should be properly quantified.
- To strengthen the demonstration that central rather than local/peripheral signals are driving the effect it would be important to show that the contralateral, uninjected or alginate only-injected, femur is also affected in wild-type mice.

Reviewer #3:

Remarks to the Author:

Metal implants have been widely used in orthopedic surgery. Magnesium cation has been shown to promote bone formation. However, the mechanism of metal cation-induced bone formation is not known. In this manuscript, authors investigated the effects of divalent metal cations including Cu, Zn and Mg on bone formation. They found that skeleton interoception regulates cation-induced bone formation. Specifically, divalent cations promote prostaglandin E₂ secretion from macrophages. This inflammation response is accompanied by the sprouting and arborization of calcitonin gene-related polypeptide- α + nerve fibers. PGE₂ activates its receptor 4 (EP4) and conveys the interoceptive signals to the central nervous system. Activating skeleton interoception downregulates sympathetic tone for new bone formation. Sensory denervation or knockout of EP4 in the sensory nerves eliminates the osteogenic effects of divalent cations. These findings provide the critical mechanism of metal cation-induced bone formation with potential application of new generation of implants. The experiments were well designed, and results are convincing. The following aspects need resolution.

Specific Comments

1. Divalent cations have in multiple prior studies in the late 1990s have been shown to affect bone resorption by osteoclasts and bone formation by osteoblasts. These references are surprisingly omitted. Likewise there is a significant body of literature from the late 1980s on the localization and release of CGRP in peripheral nerve terminals and its action on bone cells, which has again not been cited. It is important that the Introduction provides a fair representation of past work, and should be re-written to highlight previous studies in context of the author's own. With that said, this paper is novel in that it provides the vital link between divalent cation sensitivity and CGRP action on bone cells.
2. Cations including Cu, Zn and Mg induced bone formation, but at different levels. Cu had the most effect on bone formation. What is the possible explanation? Please refer, again, to early studies in osteoclasts for relative effects of divalent cations. What is the likely mechanism of action--is the calcium sensing receptor activated?
3. H type vessels are coupled with new bone formation. One would assume that angiogenesis should come along with cation-induced bone formation. There are at least two Nature papers that need consideration in this context.
4. In Fig. 2a, CD68 staining in Zn-Alg group has a considerable non-specific background. images are needed. The images of each marker should be separated.
5. In Fig. 4a, the co-staining CGRP and PGP9.5 is not clear. Better images are required.

We wanted to thank the reviewers for their thoughtful, constructive and accurate comments for our manuscript. We have addressed all the questions and comments brought forth through additional experimentation and clarification. We have highlighted all changes in the revised manuscript.

Reviewer #1 (Remarks to the Author):

Review Manuscript NCOMMS-21-28944

Divalent Metal Cations Stimulate Skeleton Interoception to Promote Bone Formation

The manuscript describes the interconnectivity of divalent cations, bone formation, macrophages, periosteum and neural signalling. There are numerous aspects that have been addressed and analysed using alginate doped with cations administered to the bone marrow cavity of mice. WT mice and several specific mouse models are used to address specific questions. This is a very elaborate and complex study.

This manuscript addresses a highly complex topic and the introductions gives the reader a concise overview of the different topics: - introducing interoception, - the importance of neural signalling for bone, - impact of prostacyclin on bone; - divalent cations in biomaterials. The authors refer to a number of previous publications from their group as background to this study – perhaps a graphical abstract of all the here introduced principles underlying the current work would help the reader to grasp the complexity without feeling compelled to study further papers for an introduction to the topic.

Response: We appreciate the Reviewer's encouraging comments. As suggested, we have drawn a graphic abstract to illustrate divalent cation stimulation of the skeleton interoception for new bone formation including the information from our new skeleton interoception publications (1, 2 below) during revision of this manuscript (Page 3). Specifically, production of PGE₂ from macrophages by the stimulation of divalent cations (e.g., Mg²⁺, Zn²⁺, and Cu²⁺) activates sensory nerve EP4 to induce phosphorylation of

CREB signaling in ventromedial hypothalamus (VMH), and the downregulation of sympathetic activity resulting in the new bone formation in periosteum. We believe this will help the readers to have a concise overview of our study.

1. Lv X, Gao F, Li TP, Xue P, Wang X, Wan M, Hu B, Chen H, Jain A, Shao Z, **Cao X**. Skeleton Interoception Regulates Bone and Fat Metabolism through Hypothalamic Neuroendocrine NPY. *Elife*. 2021 Sep 1;10:e70324. doi: 10.7554/eLife.70324
2. Xue P, Wang S, Lyu X, Wan M, Li X, Ma L, Ford NC, Li Y, Guan Y, Ding W, **Cao X**. PGE2/EP4 Skeleton Interoception Maintained by Low-Dose Celecoxib Reduces Vertebral Endplate Porosity and Spinal Pain. *Bone Res*. Bone Res. 2021 Aug 2;9(1):36. doi: 10.1038/s41413-021-00155-z.PMID: 343347922021 9(1) 1-14

The paragraph on the systemic effects on the brain resulting from the cations delivered locally raises the question whether the effects seen in the injured bone are local or if the bone homeostasis per se is altered in these animals. This raises the question of the dosage of cations applied in this experiments with respect to gaining better bone healing – how could that be translated towards a clinical application?

Response: Thank you for the valuable comments. To confirm whether the effects of the divalent cations on periosteal new bone formation is local or systematic, we generated sensory denervated mice ($Trk_{A\text{vii}}^{-/-}$) and demonstrated that sensory afferents from the injured site is indispensable in Mg^{2+} -induced phosphorylation of CREB in VMH and new bone formation in periosteum. Moreover, conditional ablation of EP4 in sensory nerve also cancelled the effects of Mg^{2+} on the activation of skeleton interoception, thus the delivery of Mg^{2+} to the injured bone failed to promote periosteal new bone formation. Collectively, our data showed the osteogenic effects of the cations seen in cortical bone are mediated by the skeleton interoceptive circuit rather than the local direct influence on bone forming cells.

To answer whether the activation of skeleton interoception contributes to systematic regulation of bone homeostasis, we also measured the bone volume, bone mineral density, cortical bone area, p -moment of inertia, and cortical bone thickness of the contralateral femurs, which were only surgically injured but not grafted with divalent cation releasing alginate. Regardless of the types of divalent cation delivered in the left femur, the cortical bone volume or density of the right femur was not significantly affected (Extended data Fig. 9). This indicates that, instead of altering the bone homeostasis systematically, the activation of skeleton interoception contributes to a precise temporal-spatial feedback to the injured site, resulting in an accurate control of bone regeneration.

The findings of our study are clinically relevant as the divalent cation crosslinked alginate used in our study replicate the ion release profile of biodegradable metal implants. For instance, the weak cross-linking potential of Mg^{2+} contributes to a quicker release of Mg^{2+} from the alginate, which matches the fast degradation rate of pure Mg implant as we reported previously (W Li, et al. *Advanced Science*. 2021). In contrast, the strong cross-linking potential of Zn^{2+} or Cu^{2+} resulted in a slower release of Zn^{2+} or Cu^{2+} from the alginate, which corresponds to the slow degradation rate of pure Zn (H Yang, et al. *Nature Communications*. 2020) or Cu implants (DM Bastidas, et al. *Corrosion Reviews*. 2019). In recent years, biodegradable metal implants have been used in orthopedic surgery and the intervention of different skeletal diseases. With controlled degradation kinetics and gradual integration with bone tissue, biodegradable metal implants have been demonstrated to be superior to traditionally used bioinert metal implants for the treatment of musculoskeletal injuries. Given the similarity in the dosage of divalent cations concerned, the findings in our study also apply to the biodegradable metal implants used clinically.

Additionally, due to the cost-effective osteogenic properties of these divalent metal cations, they are extensively used for the modification of various orthopedic biomaterials (e.g., hydrogel, biopolymer, and bioceramics). The skeleton interoceptive circuit revealed in our study also contribute to the development of novel biomaterials to elicit the therapeutic power of these divalent cations.

The connection between the cations and the macrophages is well founded by the presented data – but are there other immune cells or other cells effected by the cations – could the authors elaborate whether they have looked into other immune cells that could be involved?

Response: As suggested by the reviewer, various immune cells, such as monocyte-macrophage, T-cells, B-cells, and dendritic cells, have been shown to participate in bone healing process. In our previous study, we have reported that the effective therapeutic window for the delivery of divalent cations to coincide with the initial phase of innate immune response, which is dominated by macrophages (W Qiao, et al. *Nature Communications*. 2021). In this study, we further demonstrated that the majority of immune cells present at the key stage of new bone formation in response to the placement of divalent cation-releasing alginate are $F4/80^+CD68^+$ macrophages rather than $CD11c^+$ dendritic cells, $CD19^+$ B-cells, or $CD3^+$ T-cells (Extended data Fig. 4).

Therefore, we hypothesize that the effects of these divalent cations on new bone formation are mediated through their immunomodulation on macrophages.

To confirm the central role of macrophage in the activation of skeleton interoception, we generated macrophages depletion mouse model by injecting diphtheria toxin in $iDTR_{LysM}^{+/-}$ mice. Our data showed the presence of other immune cells could barely compensate the loss of macrophages at the early stage of bone healing process, thus, the delivery of divalent cation failed to contribute to the activation of skeleton interoception and periosteal new bone formation.

The reported effect on the nervous system is raising the question whether the supply of these ions effects signalling pathways per se as they are known as essential co-factors important for enzyme functions. Did the authors look into systemic effects in the mice following the cation administration?

Response: The divalent cations, including Mg^{2+} , Zn^{2+} , and Cu^{2+} , are widely known for their roles as catalytic and structural co-factors of many essential enzymes. Therefore, it is necessary to ensure that the placement of divalent cation-releasing biomaterials does not lead to alteration in the systemic level of these cations and the enzyme functions associated. In our previous study, we have demonstrated that the plasma Mg^{2+} concentration would not be altered by the injection of Mg-crosslinked alginate in the femur (W Qiao, et al. Nature Communications. 2021). It is either unlikely for Cu-

crosslinked or Zn-crosslinked alginate to cause any increase in the systemic Zn^{2+} or Cu^{2+} level, because both Zn^{2+} and Cu^{2+} are stronger crosslinkers for alginate so the release of these two cations are even slower than Mg^{2+} .

Additionally, the intracellular concentrations of these divalent cations are maintained far from electrochemical equilibrium through sophisticated regulation by ion transporters located on cell membranes and cellular compartments. Therefore, a mild increase in extracellular concentrations of these divalent cations won't directly lead to significant increase in the intracellular levels of these divalent cations (Maret W. *Advances in nutrition*. 2013; Andrea M.P. Romani, *Archives of Biochemistry and Biophysics*, 2011;

Edward D. Harris, Nutrition Reviews, 2001). To confirm that, we analyzed the spleen, liver, kidney, and heart tissues at both the early (week 1) and later stage (week 4) of bone healing process after the placement of divalent cation-releasing alginate. We showed that the delivery of Mg^{2+} , Zn^{2+} , or Cu^{2+} did not cause any histological alteration in these tissues, indicating their functions were not affected by the delivery of the divalent cations in the femurs (Extended data Fig. 3).

Within the manuscript the authors mention immunomodulation – but they only elaborate on PGE (as a pro-inflammatory signal) and on macrophages (as immune cells) – therefore only a very specific part of the immune reaction is being analysed – no further mention is made on the differences of the cations on the immune reaction following bone injury per se and their effect on the signalling cascade in bone is included in the manuscript. Please consider reformulating this aspect in the manuscript.

Response: Immune cells, including neutrophils, macrophages, dendritic cells, T cells, B cells, and mast cells, have been known to regulate osteoclastogenesis and osteogenesis through the secretion of inflammatory cytokines shared by the immune and skeletal systems (H Takayanagi. Nature Reviews Immunology. 2007; Z Chen, et al. Materials Today. 2016). Owing to the central role of macrophages in the immune reaction to bone biomaterials, as well as their heterogeneity and plasticity, macrophages are recognized as one of the most important target cells for immunomodulation in the biomaterial field (R.J. Miron, et al. Biomaterials. 2016). In our previous (W Qiao, et al. Nature Communications. 2021) and present study, we found that the effective therapeutic window for the delivery of divalent cations coincides with the initial phase of innate immune response, which is dominated by macrophages. Therefore, we tested our hypothesis by studying the osteogenic effects of divalent cations in macrophage depleted animal models and showed macrophage, among all the immune cells, to play a central role in the immune-neural axis.

Given their high plasticity, macrophages are well-known to play indispensable regulatory roles in bone regeneration through the production of a variety of pro-inflammatory or anti-inflammatory cytokines. Many of the inflammatory cytokines, such as IL-1 β , IL-6, and TNF- α , can be recognized by their receptors expressed in the sensory nerve to modulate the nervous system (A.F. Salvador, et al. Nature Reviews Immunology. 2021). However, PGE₂, among all inflammatory mediators, tends to receive the most attention due to its role in mediating peripheral pain pathway (Yongwoo Jang, et al. Journal of Neuroinflammation. 2020). In our previous studies, we demonstrated that PGE₂ in skeletal tissue serves as an ascending interoceptive signal to regulate bone remodeling via sympathetic nerves as the descending interoceptive pathway (H Chen, et al. Nature Communications. 2019). We further found that the PGE₂ contributes to regulation of bone homeostasis in physiological and pathological conditions by serving as a neural mediator in the skeleton interoceptive circuit (X Lv, et al. J. Clin Invest, 2020; P Xue, et al. Bone Research, 2021). Therefore, we hypothesized PGE₂ is the key inflammatory signal in skeleton interoception during bone healing process.

To confirm this observation, we generated COX2_{LysM}^{-/-} mice in which the production of PGE2 from macrophage were specifically ablated. Our data showed that the delivery of divalent cation failed to promote periosteal new bone formation via the trigger of the skeleton interoception. Therefore, given the versatile immunomodulatory effects of the divalent cations, the immune-neural axis involved in divalent cation-induced periosteal new bone formation is primarily mediated by macrophage-derived PGE2.

We agree with the reviewer that the involvement of various immune cells and the functions of the pro-/anti-inflammatory cytokines they produce need to be properly acknowledged. So, we have revised the manuscript accordingly to include the background on the immunomodulatory roles of these immune cells and cytokines, as well as the justification why we focus on macrophage-derived PGE2 in the current study.

This also is true for biomaterials that contain cations that are released over time – are they really immunomodulatory implants – or is this more of a side effect? And there is again the question of the dosage – how is a cation release from implants comparable to the dosages used in this experimental setting.

Response: We agree with the reviewer that the immunomodulatory effects of these divalent cations are highly concentration-dependent. For example, our team has demonstrated that, Mg²⁺, which has been traditionally recognized as an anti-inflammatory agent, can lead to the production of pro-inflammatory molecules, such as interleukine-8, and chemokine (C-C motif) ligand 5 (W Qiao, et al. Nature Communications. 2021), when delivered at a high dosage. To correlate our findings using Mg²⁺-releasing hydrogel with metal implants used in orthopedic surgery, we determined the Mg²⁺ release kinetics in vitro and in vivo over a period of 196 days (W Li., Advanced Science, 2021). Our data revealed the biodegradation of Mg implant starts with a rapid biocorrosion, leading to a burst release of Mg²⁺ comparable to the dosages used in this study. Therefore, the findings of this study represent the immunomodulatory effect of the Mg implant. In contrast, the degradation rates of pure Zn (H Yang, et al. Nature Communications. 2020) or Cu implants (DM Bastidas, et al. Corrosion Reviews. 2019) have been shown to be much slower than that of Mg. Thus, a slower release of Zn²⁺ and Cu²⁺ achieved with Zn-crosslinked alginate and Cu-crosslinked alginate similarly reflect the degradation behavior of the implants. We have revised the manuscript accordingly (Page 18, line 325) to include this information to help the readers understand the clinical relevance of our findings.

Minor remarks:

For the test of injecting alginate and alginate with cations into the mouse femur, please include information on the volume of alginate that was injected. Please state how the concentration of cations (10%) was chosen – as the biocompatibility of the ions is varying (as shown in extended Fig1b) – please indicate why no evaluation of the best concentration for the optimal effect was necessary for the in vivo procedure.

Response: Thank you for the valuable suggestion. A total of 0.01 mL of alginate was injected in the mouse femurs. We have included the information in the method of our manuscript (Page 24, Line 458). The rationales for the use of 10% magnesium chloride, zinc chloride and copper chloride are based on our cumulative ion release test, biocompatibility test, and ELISA test. We found that alginate treated with 10% magnesium chloride, which is a weak cross-linker, contributes to a substantial initial release of Mg^{2+} , leading to the production of PGE_2 from macrophage. Meanwhile, alginate treated with 10% zinc chloride or copper chloride, which are both strong cross-linker, only results in very mild release of Zn^{2+} and Cu^{2+} to upregulate PGE_2 without inducing cytotoxicity.

Over the past years, our team has been working on the development of metal implants and metal ion modified biomaterials for orthopaedic surgery. In addition to the cell viability test result shown in extended Fig.1b, we have also evaluated the biocompatibility of Mg^{2+} , Cu^{2+} , and Zn^{2+} using different cell types to seek for the optimal concentrations of these divalent cations. For example, our biocompatibility assays using mesenchymal stem cells (MSCs) showed the optimal concentration of these divalent cations for MSCs differ from each other greatly. Moreover, the optimal ion concentration for various types of cells can be quite different. Therefore, our future plan is to further optimize the dosage of these divalent cations in vivo to achieve the best clinical outcome with minimal side-effect.

The cation release kinetic showed that earlier time points coincide with the highest release – why did the authors chose to evaluate changes in bone after 4 weeks – how are they correlating the changes with the ion release and exclude secondary effects from participating in the resulting changes (eg. changes in the vasculature...) – in this respect please check the last sentence of the first results paragraph –“our results reveal that divalent.....”

Response: Compared with the control group (pure alginate), the periosteal new bone formation triggered by the divalent cations, including Mg^{2+} , Zn^{2+} , and Cu^{2+} , can be evident even at the early stage (week 1) after the placement of the biomaterial (Extended data Fig.4a). However, these newly formed bone tissues are yet fully mineralized. Indeed, these unstable woven bones are being actively remodeled through the interoceptive circuit resulting from the ongoing release of divalent cations. Thus, we analyzed the bone volume and density at week 4 after the operation when the newly formed cortical bone is mineralized and stable.

Nevertheless, it is difficult to exclude the involvement of vasculature because angiogenesis and vasculogenesis are vital biological procedures in new bone formation. Therefore, difference in the vascularization between the divalent cation-treated group and the control group is expected. Indeed, according to some recent studies, vasculature may be one part of the skeleton interoception through which nerve system regulates new bone formation. For example, the neuron-derived calcitonin gene-related peptide (CGRP) is found to be indispensable for type-H vessel formation, which is a biological event coupling angiogenesis and osteogenesis after bone injury (J Mi, et al. Advanced Science, 2021). Therefore, it would be both important and interesting to further elucidate role of vasculature in the skeleton interoceptive circuit. We have included it in the discussion of our manuscript (Page 21, Line 389).

Fig 2: d) PGE concentration, please indicate where this is measured?

Thank you for the question. PGE₂ was measured in the serum sample harvested from mice at week 1 or week 4 after the operation. In brief, whole blood samples were collected by cardiac puncture immediately after the mice were euthanized. Serum was collected by centrifuging the blood at 2,000 rpm for 15 minutes. We have also ensured this is properly addressed in both the figure legend and method of our manuscript.

Fig 4: a and c – the images are so small, that it is difficult to “see” the nerve fibers and their connection with CD68 + cells – could they be enlarged?

Thank you for the suggestion. We realize the quality of the image was greatly compromised after the compression. Thus, to better demonstrate the relationship between the nerve fibers and the macrophages, images captured at higher resolution was provided. Meanwhile, we further provide corresponding higher magnification of the image to show the connection between the nerve fiber and the macrophages.

Please check the following sentence: “Interestingly, CGRP+ dendrite sprouting located primarily around COX2+ macrophages in the periosteum of COX2wt mice treated with divalent cations was significantly reduced in COX2LysM^{-/-} mice (Extended data Fig. 5a). Line 177.

Response: Thank you for pointing out the issue. To express our idea more clearly, we have revised this sentence into “Interestingly, CGRP+ dendrite sprouting located primarily around COX2+ macrophages in the periosteum of COX2wt mice treated with divalent cations was missing from COX2LysM^{-/-} mice”.

Reviewer #2 (Remarks to the Author):

This manuscript aims at defining the mechanism underlying a positive activity of metallic divalent cations on bone mass in a murine injury model. It shows that the presence of these ions triggers the synthesis of PGE2 by bone marrow macrophages and an increase in CGRP levels in bone of wild-type but not Cox-2 knockout mice. It then uses sensory nerves depletion of TrkA and EP4 to show that these structures are relaying the signal to the hypothalamus to impact the sympathetic regulation of bone remodeling. This is a study that address a poorly understood observation in bone biology and it should be of general interest. The paper is generally well written but there are some technical concerns, outlined below, that should be carefully addressed before publication.

– **The title should specify that the findings relate to an injury model in the mouse.**

Response: Thank you for the suggestion. The title of our article has been revised to “Divalent Metal Cations Stimulate Skeleton Interoception for New Bone Formation in Mouse Injury Models”

– **To demonstrate de facto that the presence of divalent cations increases bone formation quantification of the bone formation rate following fluorescent labeling should be shown at least in figures 1, 3.**

Response: Thank you for the suggestion. Fluorochrome labeling has been widely used for the study on bone formation and bone remodeling dynamics, so it would be very helpful to include this part of data in our study. To better demonstrate the periosteal new bone formation rate, two fluorochrome labels were used sequentially. In brief, calcein green (5 mg/kg, Sigma-Aldrich) was subcutaneously injected into mouse femurs one week after the surgery, while xylenol orange (90 mg/kg, Sigma-Aldrich) was injected two weeks after the surgery. The intensity of fluorescence and the distance between two labeling were measured and analyzed by ImageJ software.

We showed that the release of divalent cations, including Mg^{2+} , Zn^{2+} , and Cu^{2+} , contributed to a significantly higher rate of mineral deposition, as manifested by a significantly increased fluorochrome (i.e., calcein and xylenol) intensity and an increased distance between the two fluorochrome labels (Fig. 1g-i, Extended data Fig. 2d).

– The quantification of osteoblast and osteoclast numbers should be shown in the main figure 1 instead of the extended data. These data should also be shown for the experiments presented in figure 3.

Response: Thank you for the suggestion. The quantifications of osteoblast and osteoclast numbers in Fig.1 and Fig.3 have been shown in main figures.

– In figure 1, 2 and extended figure 3 the immunofluorescence data have not been quantified, only representative fields are shown. Unless a proper quantification is performed, the corresponding text should not refer to “significant” changes in expression.

Response: Thank you for the suggestion. We agree that a proper quantification of the immunofluorescence data is essential for the interpretation of the data, thus, we have included the quantification of Fig.1d and Fig.2a in Fig. 1f and Fig. 2b, respectively. The immunofluorescent images in extended figure 3 (extended Fig.5 in revised version) showed the number of CD68+ macrophage and expression of COX2, which was quantified in Fig. 2b.

- At least some of the *in vitro* data shown in figure 2f-h should be confirmed using cells directly isolated from the implanted mice.

Response: To verify our *in vitro* finding, we analyzed the gene expression of *PTGES* and *COX2* in macrophages isolated from the mouse femurs grafted with divalent cation releasing alginate. In brief, we generated LysMCre-RosaYFP mouse in which macrophages carry YFP fluorescence. After the placement of Mg²⁺, Zn²⁺, or Cu²⁺ releasing alginate, we used flow cytometry sorter to isolate the YFP⁺ macrophage and extract the RNA for RT-qPCR analysis. We found that the delivery of divalent cations contributed to a significant upregulation in the expression of *PTGES* and *COX2*. This data further confirmed our finding and is shown in Fig. 2h.

- The western blots for changes in CREB/P-CREB and HT2C hypothalamic levels should be shown in figure 5. The number of mice used for these experiment should be clarified.

Response: As requested, we have moved the western blots and corresponding quantification to Fig.5c, d. The number of mice used for the experiment (5) has been clarified in the legend.

- The changes in CREB/P-CREB signal shown in figure 5a are not convincing. This should be properly quantified.

Response: Thank you for the suggestion. The phosphorylation of CREB in the VMH has been quantified by the ratio (%) of phosphorylated CREB to total CREB in Fig. 5b.

- To strengthen the demonstration that central rather than local/peripheral signals

are driving the effect it would be important to show that the contralateral, uninjected or alginate only-injected, femur is also affected in wild-type mice.

Response: Thank you for the valuable comments from the reviewer. To answer whether the activation of skeleton interoception contributes to central regulation of bone homeostasis that affect the contralateral femurs, we placed the divalent cation-releasing alginate in the left femur and left the contralateral (right) femur only surgically injured but not grafted with anything. Our data showed that, regardless of the types of divalent cation delivered in the left femur, the bone volume, bone mineral density, cortical bone area, ρ -moment of inertia, and cortical bone thickness of the contralateral (right) femur remained unchanged. This indicates that, instead of altering the bone homeostasis systematically, the activation of skeleton interoception contributes to a precise temporal-spatial feedback to the injured site through sympathetic nerve system.

To confirm this, we generated sensory denervation ($\text{Trk}_{\text{A}v\text{i}l}^{-/-}$) mouse model and demonstrated that sensory afferents from the injured site is indispensable in Mg^{2+} -induced phosphorylation of CREB in VMH and new bone formation in periosteum. Moreover, conditional ablation of EP4 in sensory nerve also cancelled the effects of Mg^{2+} on the activation of skeleton interoception. In this case, the delivery of Mg^{2+} to the injured bone failed to promote periosteal new bone formation. Collectively, our data showed the osteogenic effects of the cations are mediated by the skeleton interoceptive circuit rather than the local direct influence on bone forming cells.

Reviewer #3 (Remarks to the Author):

Metal implants have been widely used in orthopedic surgery. Magnesium cation has been shown to promote bone formation. However, the mechanism of metal cation-induced bone formation is not known. In this manuscript, authors investigated the effects of divalent metal cations including Cu, Zn and Mg on bone formation. They found that skeleton interoception regulates cation-induced bone formation. Specifically, divalent cations promote prostaglandin E2 secretion from macrophages. This inflammation response is accompanied by the sprouting and arborization of calcitonin gene-related polypeptide- α nerve fibers. PGE2 activates its receptor 4 (EP4) and conveys the interoceptive signals to the central nervous system. Activating skeleton interoception downregulates sympathetic tone for new bone formation. Sensory denervation or knockout of EP4 in the sensory nerves eliminates the osteogenic effects of divalent cations. These findings provide the critical mechanism of metal cation-induced bone formation with potential application of new generation of implants. The experiments were well designed, and results are convincing. The following aspects need resolution.

We appreciate the reviewer's encouraging comments.

Specific Comments

1. Divalent cations have in multiple prior studies in the late 1990s have been shown to affect bone resorption by osteoclasts and bone formation by osteoblasts. These references are surprisingly omitted. Likewise there is a significant body of literature from the late 1980s on the localization and release of CGRP in peripheral nerve terminals and its action on bone cells, which has again not been cited. It is important that the Introduction provides a fair representation of past work, and should be re-written to highlight previous studies in context of the author's own. With that said, this paper is novel in that it provides the vital link between divalent cation sensitivity and CGRP action on bone cells.

Response: Thank you for the valuable comments. We believe it is both important and essential to acknowledge the early studies on the osteogenic effects of divalent cations and the function of CGRP on bone cells. Therefore, we have revised the introduction and discussion of our manuscript to include the literatures mentioned:

Page 5 Line 14:

“It has been known from the late 1990s that various divalent metal cations, such as magnesium ions (Mg^{2+}), zinc ions (Zn^{2+}), and copper ions (Cu^{2+}), play vital roles in bone growth, modeling, and remodeling¹⁻³. Over the decades, the regulatory effects of these divalent cations on osteogenesis, osteoclastogenesis, angiogenesis, and immune responses have been gradually revealed⁴⁻⁹. However, it was not until recently that the involvement of nervous system in the new bone formation induced by divalent metal cations has begun to be realized¹⁰.”

Page 19 Line 353:

“In addition to its major role as a neurotransmitter and neuromodulator, CGRP is also considered a peptide that can be released from the peripheral nerve terminals to regulate osteoclast and osteoblasts^{68,69}, which was recently reported to be implicated in the new bone formation induced by pure magnesium implant¹⁰. In our study, a conspicuous increase in the number of CGRP+ nerve fibers in divalent cation–treated femurs suggests an association between the activation of sensory afferents and the inflammatory microenvironment, which is indispensable in skeleton interoceptive circuit. Therefore, the upregulation of CGRP in response to divalent cations may possess multiple functions, including the nociceptive transmission contributing to central sensitization and the direct control of bone cells through the receptors they shared.”

2. Cations including Cu, Zn and Mg induced bone formation, but at different levels. Cu had the most effect on bone formation. What is the possible explanation? Please refer, again, to early studies in osteoclasts for relative effects of divalent cations. What is the likely mechanism of action--is the calcium sensing receptor activated?

Response: Thank you for the valuable comments. Although we observed Cu^{2+} , Zn^{2+} and Mg^{2+} all contribute to new bone formation through the activation of interoceptive circuit, the bone formation outcome can differ due to several other factors. For example, as suggested by the reviewer, Cu^{2+} has been known to have a substantial influence on osteoclastogenesis (T Wilson, et al. Calcified tissue international. 1981; L Yang, et al. Biomaterials, 2010; A Bernhardt, et al. Int J Mol Sci, 2021). This resulted in the failed resorption of old bone from endosteal surface, thus, the morphology of Cu-Alg grafted femur is quite different from the others. We have revised our manuscript to include the information in the discussion.

Page 20 Line 378:

“The discovery of the skeleton interoception-mediated new bone formation triggered by Mg^{2+} , Zn^{2+} , and Cu^{2+} indicates that the complexity of the underlying mechanism for the osteogenic effects of these divalent cations has been greatly underestimated.

Nevertheless, it is important to note that there have been multiple approaches reported in the past few decades through which these divalent cations may modulate bone homeostasis⁶⁻⁹. And this may explain the difference among the three tested divalent cations in term of their bone formation outcome. For instance, in our study, the highest bone volume was observed in the Cu-Alg group, as the old bone was barely resorbed from the endosteal surface by osteoclasts following the new bone formation on periosteal surface. In fact, from as early as 1981, Cu²⁺ has been reported to have a direct dose-dependent inhibitory effects on osteoclastic activity⁷⁴. Cu²⁺ has also been used in biomaterials to shift the equilibrium between bone formation and bone resorption, because the tolerance of osteoclasts and osteoblasts to Cu²⁺ differs^{75,76}.”

3. H type vessels are coupled with new bone formation. One would assume that angiogenesis should come along with cation-induced bone formation. There are at least two Nature papers that need consideration in this context.

Response: Thank you for the professional comments. Angiogenesis and vasculogenesis are indeed vital biological procedures in new bone formation. According to some recent studies, vasculature may be one part of the skeleton interoception through which nerve system regulates new bone formation. Therefore, we have revised our manuscript to include it in the discussion.

Page 21 Line 389:

“In addition to the bone cells, some divalent cations like Mg²⁺ and Zn²⁺ have also been shown to facilitate type H vessel formation by targeting endothelial cells, which couples angiogenesis with osteogenesis during bone healing⁶. This subtype of vessels, characterized by the co-expression of CD31 and endomucin, can provide niche signals for perivascular osteoprogenitors to promote osteogenesis^{77,78}. Recent studies showed the sensory innervation following bone injury is an essential upstream mediator for vasculature⁴⁵, and electrical stimulation at DRG could promote type-H vessel formation to enhance bone regeneration⁷⁹. Therefore, it would be interesting to further explore to which extend are angiogenesis and vasculogenesis involved in skeleton interoception and what their specific role may be in divalent cation-induced new bone formation.”

4. In Fig. 2a, CD68 staining in Zn-Alg group has a considerable non-specific background. images are needed. The images of each marker should be separated.

Response: Thank you for the suggestion. We recaptured the immunofluorescent images for Zn-Alg group to better localize the CD68+ macrophage in periosteum after the operation. Additionally, as suggested by the reviewer, we modified the images by providing both the merged images and the images in single channels (i.e., CD68 and COX2) in Fig.2a.

5. In Fig. 4a, the co-staining CGRP and PGP9.5 is not clear. Better images are required.

Response: Thank you for the suggestion. We realize the quality of the image was greatly compromised after the compression. Thus, to better demonstrate the morphology of the nerve fibers, images captured at higher resolution and corresponding higher magnification of the area of interest are provided in Fig. 4a.

Reviewers' Comments:

Reviewer #1:

Remarks to the Author:

All review comments were adequately addressed.

Reviewer #2:

Remarks to the Author:

My comments have been addressed.

Reviewer #3:

Remarks to the Author:

The authors have thoughtfully and thoroughly responded to review concerns. There are no outstanding issues.